## BRIEF REPORT

# Distinct evolutionary trajectories of subgenomic centromeres in polyploid wheat

Yuhong Huang[1†], Yang Liu[1†], Chang Liu[1†], Congyang Yi[1], Jinsheng Lai[2], Hongqing Ling[1], Handong Su[3*] and Fangpu Han[1*]

†Yuhong Huang, Yang Liu and Chang Liu contributed equally to this work.

*Correspondence:
shandong@mail.hzau.edu.cn;
fphan@genetics.ac.cn

[1] Institute of Genetics and Developmental Biology, Chinese Academy of Sciences, 100101 Beijing, China
[2] China Agricultural University, Beijing 100193, China
[3] National Key Laboratory of Crop Genetic Improvement, Hubei Hongshan Laboratory, Shenzhen Institute of Nutrition and Health, Huazhong Agricultural University, Wuhan 430070, China

## Abstract

**Background:** Centromeres are crucial for precise chromosome segregation and maintaining genome stability during cell division. However, their evolutionary dynamics, particularly in polyploid organisms with complex genomic architectures, remain largely enigmatic. Allopolyploid wheat, with its well-defined hierarchical ploidy series and recent polyploidization history, serves as an excellent model to explore centromere evolution.

**Results:** In this study, we perform a systematic comparative analysis of centromeres in common wheat and its corresponding ancestral species, utilizing the latest comprehensive reference genome assembly available. Our findings reveal that wheat centromeres predominantly consist of five types of centromeric-specific retrotransposon elements (CRWs), with CRW1 and CRW2 being the most prevalent. We identify distinct evolutionary trajectories in the functional centromeres of each subgenome, characterized by variations in copy number, insertion age, and CRW composition. By utilizing CENH3-ChIP data across various ploidy levels, we uncover a series of CRW invasion events that have shaped the evolution of AA subgenome centromeres. Conversely, the evolutionary process of the DD subgenome centromeres involves their expansion from diploid to hexaploid wheat, facilitating adaptation to a larger genomic context. Integration of complete einkorn centromere assemblies and *Aegilops tauschii* pangenomes further revealed subgenome-specific centromere evolutionary trajectories. By inclusion of synthetic hexaploid from $S_2$-$S_3$ generations, alongside $2x/6\times$ natural accessions, we demonstrate that DD subgenome centromere expansion represents a gradual evolutionary process rather than an immediate response to polyploidization.

**Conclusions:** Our study provides a comprehensive landscape of centromere adaptation, evolution, and maturation, along with insights into how retrotransposon invasions drive centromere evolution in polyploid wheat.

**Keywords:** *Triticum aestivum*, FILTR-RTs, Centromere evolution, CENH3, CRW

## Background

As a major food crop, common wheat ($2n = 6x = 42$; BBAADD) played a significant role in human civilization and development [1]. Wheat evolution starts 6.5–7.0 million years ago (Ma) from a progenitor that gave rise to the A, B, and D lineages that merged to form common wheat [2–6]. The A lineage gave rise to *Triticum urartu* ($2n = 2x = 14$; AA) and the B lineage likely arose from an extinct or unknown *Aegilops* species rather than a mosaic genome [7–9]. Following genome divergence, an ancient homoploid hybrid occurred ~5.5 Ma between B lineage and A lineage, ushering in the first merge event and forming *Ae tauschii* ($2n = 2x = 14$; DD) [2, 4]. These three evolved in parallel until the A and B genomes reunited to form an allotetraploid species *T. turgidum* ssp. *dicoccoides* ($2n = 4x = 28$; BBAA) [3, 7]. Thus, the centromeres of the AA and BB Subgenomes were wrapped in the same nuclear envelope and continued to coevolve for 800,000 years. *T. turgidum* then merged with *Ae. tauschii*, whose chromosomes are notably smaller than those of the A/B genomes, to form the modern common wheat fewer than 10,000 years ago [4, 5, 7, 10–12]. As a result, allohexaploid common wheat, bearing three groups of genetically divergent centromeres at different stages of polyploidy evolution, is an ideal system for studying genome evolutionary biology.

The centromere is present in each chromosome and plays a crucial role in assembling kinetochores during cell division, enabling the chromosomes to be pulled to the poles of dividing cells [13–16]. Although the centromere has a highly conserved function, it is one of the fastest evolving and most diverse regions of the genome [17–19]. To investigate centromere evolution, it is necessary to accurately identify these repeated sequences one by one in chronological order. Several mechanisms have been proposed for the formation of CENH3 binding to core younger satellite with less accumulated mutations, of which satellite homogenization and retrotransposon-driven diversification cycles [20, 21] and layered expansions [22] are two leading models in *Arabidopsis thaliana* and human, respectively. However, these models are based on satellite-rich centromeres, and the evolutionary trajectory of retrotransposon (RT)-rich centromeres remains unclear. Long terminal repeat-retrotransposons (LTR-RTs) serve as molecular "fossils," providing a temporal record of centromere evolution [23–26]. The age of LTR-RT provides a representation of centromere evolution and avoids the daunting challenges encountered in the alignment of highly repetitive sequences. Recent work implicates specific transposable elements in diploid einkorn wheat, suggesting that LTR elements facilitate new CENH3 deposition through recruitment and/or phasing, thereby highlighting a dynamic interplay between TEs and the host plant in centromere maintenance [27]. However, the mechanisms by which RTs affect centromere function and evolution in polyploid wheat remain largely unexplored.

Early studies on the genetics of functional centromeres in wheat were generally based on limited BAC sequences, focusing on specific types of RTs, and ending with static descriptive analyses which limited our understanding [28, 29]. Despite various versions of the reference genome being available for years, current genome assembly strategies resulted in varying extent of collapse in the wheat (peri)centromere [12, 30–35]. Recently, a study has shown that CRW and *Quinta* densities in centromeric regions of sequenced wheat species increase progressively from diploid to tetraploid and hexaploid stages [36]. However, the genomes used in these studies

contain numerous unassembled sequences, particularly in centromeric regions with substantial gaps, which may impact the analysis results. Specifically, early results were mainly investigated by cytological analysis, but cytological analysis methods could not distinguish repeats with similarity higher than a certain threshold. The combination of incomplete centromeric region assembly in reference genomes and prior cytological analyses with limited resolution inevitably blurred some results.

With advances in the latest near-complete reference genome assembly of *T. aestivum* cv Chinese Spring (CS) (CS-CAU) [37], we integrated CENH3-ChIP datasets from hexaploid wheat, its tetraploid and diploid relatives, and existing reference data from domesticated and wild einkorn, as well as the pan-genome of *Ae. tauschii* [35, 38]. These comprehensive datasets allowed us to redraw the sophisticated evolutionary scenarios of wheat centromeres. The elaborate full-length LTR-retrotransposons (flLTR-RTs) landscape of wheat centromeres allows us to fully decode RTs classification and distribution and trace the evolution of centromere RTs at the molecular level of AA and DD subgenomes during the recursive polyploidization.

## Results

### The centromere landscape in common wheat

To investigate the centromere landscape in common wheat with near-complete genome assembly (CS-CAU), we performed ChIP-seq with antibodies specific to the centromeric histone variant CENH3 [39]. We delimited the relatively enriched regions as the functional centromeres by mapping the anti-CENH3 ChIP-seq and Input-seq datasets to CS-CAU, respectively (Fig. 1A–C). The centromere sizes in the AA Subgenome range from 5.6 to 7.8 Mb, with an average of 6.5 Mb. In the BB Subgenomes, the sizes range from 5.0 to 7.3 Mb, averaging 5.7 Mb, while in the DD Subgenome, they range from 5.0 to 7.2 Mb, with an average of 6.0 Mb (Table 1). Centromere sizes in this system do not follow the subgenome size hierarchy (BB > AA > DD). This might be due to our limited samples of only three subgenomes, which does not invalidate the broader pattern across eukaryotes [40].

To comprehensively characterize the composition of centromeres in common wheat, we employed RepeatExplorer [41] and LASTZ [42] software to annotate the centromere sequences. Our analysis revealed that wheat centromeres are predominantly composed of long terminal repeat retrotransposons (LTR-RTs), with a smaller proportion of satellite sequences, which range from 0 to 5% (Additional file 1: Fig. S1 and S2). The satellite sequences have unit sizes of 566 and 550 bp, consistent with the previously reported CentT566 and CentT550 [11]. CentT566 is primarily distributed in BB subgenome, while CentT550 is predominantly found in DD subgenome (Additional file 1: Fig. S1A). Furthermore, the genomic landscape of centromeres varies significantly between the subgenomes (Fig. 1D; Additional file 1: Fig. S2). Notably, the centromeric sequences in the DD subgenome show a clear divergence from those found in the AA and BB subgenomes. Each centromere displays unique sequence and structural characteristics on its respective chromosome, along with distinct CENH3 profiles (Fig. 1A–C; Additional file 1: Fig. S2).

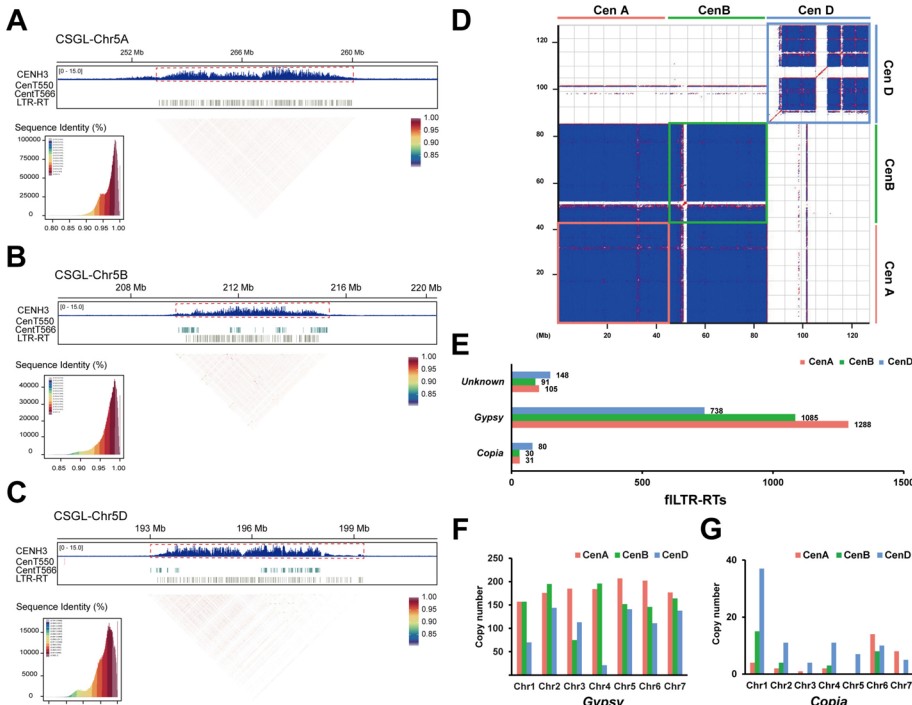

**Fig. 1** Fine structure of centromere repeat arrays in hexaploid wheat *T. aestivum*. Characteristics of Cen5A (**A**), Cen5B (**B**), and Cen5D (**C**) are shown. Different layers display the CENH3 enrichment [$\log_2$(ChIP/Input)], centromeric satellites centT550 and CentT566 distribution, centromeric flLTR-RTs annotations, and a heatmap of pairwise satellite sequence similarity. **D** Dot plots comparing the centromeres from AA, BB, and DD Subgenome assembly using a 500-bp search window. Red, green, and blue lines/boxes represent the centromeric regions from the AA, BB, and DD subgenomes, respectively. Red and blue points indicate forward- and reverse-strand similarity. **E** Distribution of LTR-RTs in the *T. aestivum* AA, BB, and DD Subgenomes across the 21 centromeres. **F** *Gypsy* Superfamily is evenly distributed across the 21 functional centromere regions. **G** *Copia* superfamily is enriched in the first homoeologous group, especially Cen1D

## Dynamics of centromere evolution in common wheat

Considering that the LTR-RTs account for the majority of wheat centromeres, we explored the global landscape of LTR-RTs in wheat centromeres and their role in centromere evolution. We employed LTR_finder [43] and LTRharvest [44] tools to identify LTR-RTs in the functional centromeres of the CS-CAU reference genome. Additionally, LTR_retriever [45] was used for integration. In total, 3595 flLTR-RTs were detected in the functional centromeres of common wheat, with 1424 in the AA, 1206 in the BB, and 965 in the DD subgenome. According to the previously defined classification criteria [46], 86.5% (3111/3595) of the candidate flLTR-RTs were annotated as belonging to the *Gypsy* Superfamily, 3.9% (141/3595) to the *Copia* Superfamily, and 9.6% (344/3595) to an unknown superfamily (Fig. 1E). The *Gypsy* Superfamily is evenly distributed across the 21 functional centromeric regions (Fig. 1F), while the *Copia* superfamily is enriched in the first homoeologous group, particularly in Cen1D (Fig. 1G). To further refine the classification of these superfamilies, we performed phylogenetic analysis on the candidate flLTR-RTs. RLG_famc8.3 (*Cereba*), RLG_famc8.2 (*Quinta*), and RLG_famc39 (*Abia*) retrotransposons were treated as references [47]. Centromere-specific RT sequences from maize [48], rice [49], and rye (*Secale cereale*) [50] were used as outgroups. The cladogram can be divided into five clades, including two *Cereba* subfamilies (named

**Table 1** Delimiting *T. aestivum* centromeres based on CENH3 ChIP-seq

| Chromosome | CENH3-enriched intervals | |
|---|---|---|
| | Pseudomolecule positions (Mb) | Total length (Mb) |
| 1A | 213.0–219.2 | 6.2 |
| 2A | 346.2–352.1 | 5.9 |
| 3A | 332.0–338.2 | 6.2 |
| 4A | 288.5–295.0 | 6.5 |
| 5A | 253.0–260.0 | 7.0 |
| 6A | 285.7–293.5 | 7.8 |
| 7A | 368–373.6 | 5.6 |
| 1B | 236.0–243.3 | 7.3 |
| 2B | 366.3–373.0 | 6.7 |
| 3B | 362.0–365.2 | 3.2 |
| 4B | 328.6–334.7 | 6.1 |
| 5B | 210.0–215.0 | 5.0 |
| 6B | 346.2–351.8 | 5.6 |
| 7B | 314.2–320.3 | 6.1 |
| 1D | 172.8–180.0 | 7.2 |
| 2D | 269.8–276.5 | 6.6 |
| 3D | 252.2–258.0 | 5.8 |
| 4D | 186.6–191.6 | 5.0 |
| 5D | 193.2–199.3 | 6.1 |
| 6D | 239.0–244.8 | 5.8 |
| 7D | 347.6–353.0 | 5.4 |

CRW1 and CRW2, respectively). A *Cereba* subfamily CRW2 and *Quinta* were mixed into one branch. We split it into two subfamilies and named *Quinta* as CRW3. A small number of *Abia* formed a separate clade, which we named CRW4. In particular, the fifth group, which is classified as CRW5, has topologically long and scattered branches with many RTs mixed (including CRM1 in maize, pAWRC1 in rye, and almost all *Copia* and unknown superfamily), suggesting that CRW5 represents a class of retrotransposons with earlier differentiation and richer heterogeneity (Fig. 2A; Additional file 1: Fig. S1B).

Since CRW1 and CRW2 are too similar to each other for cytological analysis to be distinguishable, they are artificially grouped together and referred to as CRW1&2. RT invasions and degradations are two opposing forces that have promoted the accumulation of CRWs in (peri)centromeres of diploid and hexaploid wheat, resulting in the predominant enrichment of CRW1&2, as well as trace amounts of CRW3 and CRW4 enrichment in the centromeric region (Additional file 1: Fig. S3). To investigate the dynamics of centromeric retrotransposons (CRWs) in the functional centromeres of common wheat, we estimated insertion times of flLTR-RTs across the A, B, and D centromeres (Fig. 2B). Our findings revealed that the D centromere (CenD) exhibited a higher proportion of aged CRWs, while there was no significant difference in the age distribution between the A (CenA) and B (CenB) centromeres (Fig. 2C). Furthermore, we observed that CRWs from CenA and CenB clustered closer, whereas those from the CenD formed a distinct cluster, indicating closer evolutionary relationship

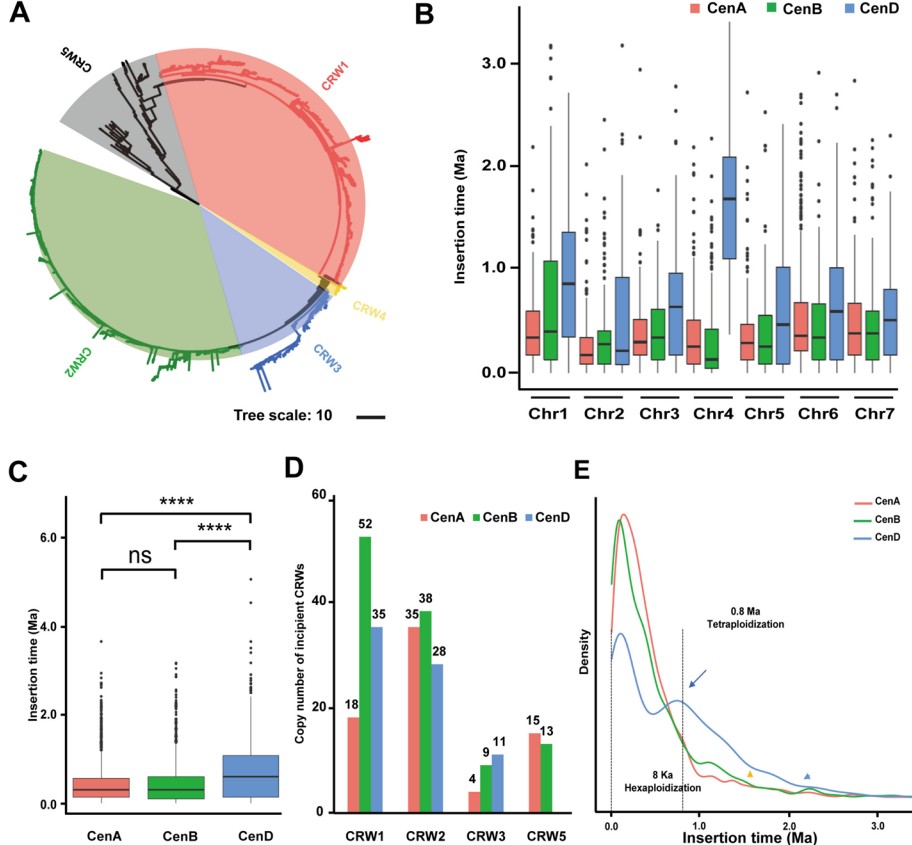

**Fig. 2** Centromeric-specific retrotransposon element (CRWs) composition and insertion history in common wheat. **A** Phylogenetic tree based on flLTR-RTs from *T. aestivum* centromeres, with centromere-specific RTs sequences from corn, rice, and rye as outgroups. Among the five CRWs specifically presenting in the functional centromeres of *T. aestivum*, CRW1 is shown in red, CRW2 is in green, CRW3 is in blue, CRW4 is in yellow, and CRW5 is in black. **B** The insertion time of CRWs on homoeologous group Chr1 to Chr7, shown as histograms. Different colors represent CRWs from different wheat sub-centromeres. **C** Older CRWs were detected in the D subgenomic centromeres compared to the A and B subgenomic centromere. Statistical analyses were completed using the Mann–Whitney *U* test. Asterisks represent statistically significant differences (*$P < 0.05$, **$P < 0.01$, ***$P < 0.001$, ****$P < 0.0001$) between the indicated groups. **D** Copy number of incipient CRWs among three sub-centromeres. **E** The insertion time of CRWs, as sampled in three sub-centromeres of *T. aestivum*. CenA in red; CenB in green; and CenD in blue. The *x*-axis represents the time (Ma) of CRW insertion into the host genome; the *y*-axis represents the density of insert time. The bottom blue triangle Marks the time when CRW burst began to be detected in CenD, with yellow Marking CenA and CenB. The second polyploid event occurred 8000 years ago (ka)

between the AA and BB subgenomes (Additional file 1: Fig. S1C). Notably, CenB had the highest amounts of incipient CRWs with the identical LTR sequences on both ends (insertion time = 0), suggesting that CRWs in CenB have been much more active than those of CenA and CenD (Fig. 2D). The density plot revealed synchronous peaks of RT burst events in CenA, CenB, and CenD, with an additional non-major peak in CenD at 0.8 Ma (blue arrow indicated in Fig. 2E, which will be discussed next). However, there is a time lag when RT burst events begin to be detected among three subcentromeres: CenD first, followed by CenA and CenB (Fig. 2E). In summary, the evolutionary trajectories of the functional centromeres in each subgenome differ significantly in terms of CRW copy number, and insertion age.

### Evolutionary scenario of the AA subgenomic centromeres

To investigate the molecular evolution of AA subgenomic centromeres in polyploid wheat, we generated multiple sets of anti-CENH3 ChIP-seq datasets from wheat lines with different ploidy levels. We further integrated some publicly available profiles across representative accessions with high-quality reference genomes, spanning diploid *T. monococcum* ($A^mA^m$) and *T. urartu* ($A^uA^u$), tetraploid *T. turgidum* (BBAA), and hexaploid *T. aestivum* lineages, as detailed in Table 2. Technical replicates were performed, demonstrating high reproducibility of the ChIP-seq data (Additional file 1: Fig. S4). We first mapped the CENH3 ChIP-seq reads from three *T. urartu* accessions (G1812, TMU06, and TMU38) and four *T. monococcum* accessions (KU104, KT003-002, TA10622, and TA299) to the *T. urartu* reference genome (G1812 accession) [33]. Continuous CENH3 peaks were detected based on the diploid *T. urartu* genome (Additional file 1: Fig. S5). We found that the CENH3 peaks at the centromeres of *T. urartu* wheat are more concentrated and contiguous, whereas the peaks in *T. monococcum* lines appear more scattered (Additional file 1: Fig. S5). Additionally, significant variations in centromeric CENH3 profiles are evident among different AA lines. These findings suggest that centromeres have undergone some degree of differentiation between $A^uA^u$ and $A^mA^m$ wheat. However, when mapped these CENH3 ChIP-seq datasets to the CS-CAU reference genome, only the datasets from the AA subgenome of tetraploid and hexaploid wheat exhibited obvious centromeric peaks (Additional file 1: Fig. S6). Unlike the

**Table 2** Species and datasets used in this study (public data and current study)

| Species and cultivars | Accession | Genome formula | Data source |
|---|---|---|---|
| **Diploid (2*n* = 2*x* = 14)** | | | |
| *T. monococcum* | KU104 [51] | $A^mA^m$ | This study |
| *T. monococcum* | KT003-002 [52] | $A^mA^m$ | This study |
| *T. monococcum* | TA10622 | $A^mA^m$ | (Ahmed et al., 2024) [35] |
| *T. monococcum* | TA299 | $A^mA^m$ | (Ahmed et al., 2024) [35] |
| *T. urartu* | G1812 [33] | $A^uA^u$ | This study |
| *T. urartu* | TMU06 [53, 54] | $A^uA^u$ | This study |
| *T. urartu* | TMU38 [53, 54] | $A^uA^u$ | This study |
| *Ae. tauschii* | Y2282 [36] | DD | This study |
| *Ae. tauschii* | TA10171 | DD | (Cavalet-Giorsa et al., 2024) [38] |
| *Ae. tauschii* | TA1675 | DD | (Cavalet-Giorsa et al., 2024) [38] |
| *Ae. tauschii* | TA2576 | DD | (Cavalet-Giorsa et al., 2024) [38] |
| *Ae. tauschii* | TQ27 [55] | DD | This study |
| *Ae. tauschii* | AS2392 [56] | DD | This study |
| *Ae. tauschii* | AS2389 [57] | DD | This study |
| **Tetraploid (2*n* = 4*x* = 28)** | | | |
| *T. turgidum* ssp. *dicoccoides* | Zavitan [3] | $BBA^uA^u$ | This study |
| *T. turgidum* ssp. *durum* | Svevo [58] | $BBA^uA^u$ | This study |
| **Hexaploid (2*n* = 6*x* = 42)** | | | |
| *T. aestivum* | CS [37] | $BBA^uA^uDD$ | This study |
| *T. aestivum* | AK58 [59] | $BBA^uA^uDD$ | (Liu et al., 2023) [60] |
| *T. aestivum* | TAA10 [61] | $BBA^uA^uDD$ | This study |
| *T. turgidum* (Zavitan) × *Ae. tauschii* (Y2282) | Za-Y2 | $BBA^uA^uDD$ | This study |

continuous CENH3 peaks observed when mapped to the *T. urartu* reference genome, the peaks appeared scattered and intermittent when mapped to the CS-CAU reference genome (Fig. 3A; Additional file 1: Fig. S6).

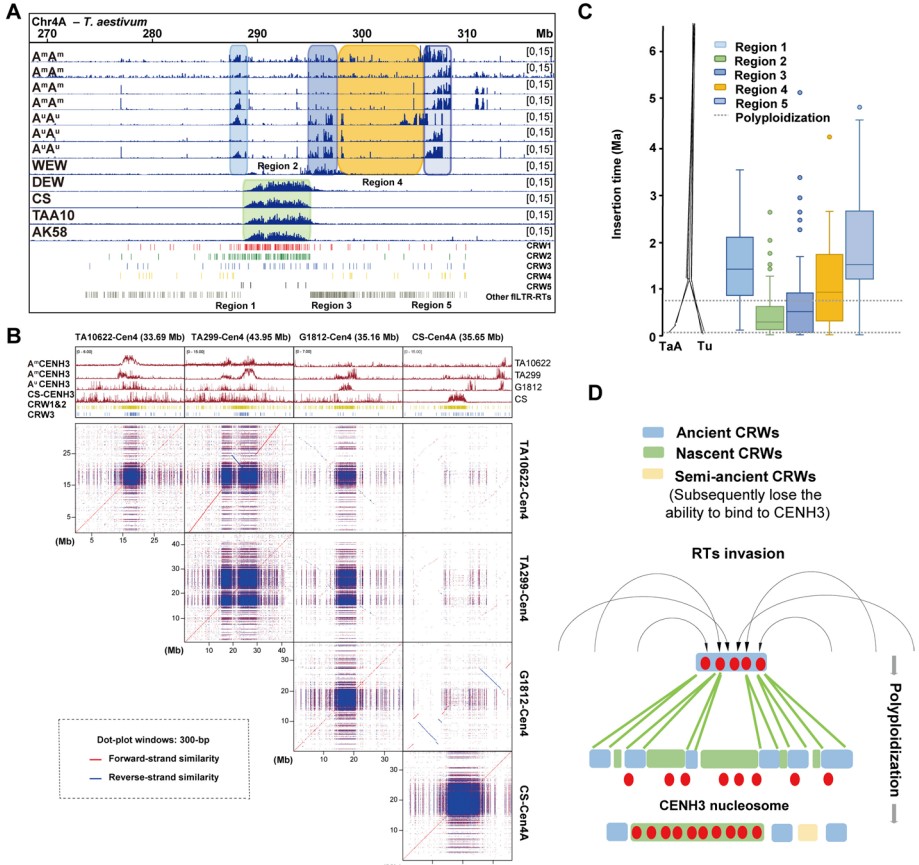

**Fig. 3** A prospective evolutionary model for common wheat CenA-CS. **A** The detailed layout of the CENH3 enrichment [log₂(ChIP/Input)] ratio along the pericentromeric region of bread wheat chromosome 4A. The first four tracks (1–4) are *T. monococcum* (AᵐAᵐ, accessions TA299, TA10622, KT003-002, and KU104). The next three (5–7) are *T. urartu* (AᵘAᵘ, accessions TMU06, G1812, and TMU38). Tracks eight and nine (8–9) are wild emmer wheat (WEW, accession Zavitan) and domesticated emmer wheat (DEW, accession Svevo), belonging to *T. turgidum* (BBAA). Tracks ten to twelve (10–12) are *T. aestivum* (accessions CS, TAA10, and Aikang58 (AK58)). The distribution of CRW1 (red), CRW2 (green), CRW3 (blue), CRW4 (yellow), CRW5 (black), and other flLTR-RTs (gray) are shown at the following tracks, respectively. **B** Dot plot alignments (300-bp window) of pericentromeric and centromeric regions for chromosomes 4 A, comparing *T. monococcum* (domesticated einkorn TA10622; wild einkorn TA299), *T. urartu* (G1812), and *T. aestivum* (CS). Genomic coordinates (top) and CENH3 ChIP-seq coverage profiles (middle) demarcate centromere boundaries. Distributions of CRWs are annotated as colored bars. Sequence similarities on forward and reverse strands are indicated in red and blue, respectively. **C** The insertion time of CRWs associated with corresponding subregions, analyzed by boxplot. Two successive polyploid events are indicated on the timescale in the left panel. The evolutionary trajectories of *T. aestivum* AA subgenome (TaA) and *T. urartu* (Tu) are shown as separate black lines. **D** A predictive evolutionary model of AA sub-centromeres in allohexaploid wheat. Episodic invasions by CRWs (black arrows) displace ancestral centromeric repeats to pericentromeric domains, driving dynamic redistribution of CENH3 nucleosomes between ancestral CRW clusters (blue shading) and emerging CRW arrays (green shading). Epigenetic competition culminates in dominant CENH3 occupancy at nascent retrotransposon arrays, while relocating ancient CRWs to unilateral or bilateral pericentromeric regions. Through continuous integration of novel CRW insertions and reorganization of ancestral repeats, this process establishes an evolutionarily dynamic CRW cluster architecture that ultimately shapes the functional centromeres of the AA subgenome

Comparative dot plot revealed that the centromere sequences of the common wheat CS-AA subgenome have undergone pronounced differentiation when compared to those of the diploid A$^u$A$^u$ and A$^m$A$^m$ genomes, and the divergence between A$^u$A$^u$ centromeres and CS-AA subgenome centromeres is comparatively less (Fig. 3B; Additional file 1: Fig. S7). Notably, the adherence capacity of CRWs within centromere is not influenced by allopolyploidization, as the content tend of CRWs in diploid AA genome, as well as hexaploid wheat, remains consistent (Additional file 1: Fig. S8A-8C). The evolutionary dynamics of these centromeres were further elucidated through phylogenetic analysis of CRWs. Notably, CRW1&2 elements showed distinct clustering patterns between diploid and hexaploid centromeres, while CRW3 maintained conserved features across ploidy levels (Additional file 1: Fig. S9). Analysis of CRW insertion chronology showed that wild einkorn *T. monococcum* centromeres contained significantly younger CRWs compared to *T. urartu* and CS-AA. Intriguingly, while chromosomes 1–5 exhibited younger insertions in CS-AA relative to *T. urartu*, chromosomes 6–7 displayed the opposite trend (Fig. S8D). These results suggest that the centromere sequences of AA subgenome have undergone substantial variation during polyploidization.

To further understand the dynamic changes of centromeres during polyploidization, we divided the scattered centromeric regions identified by mapping *T. urartu*'s and *T. monococcum*'s CENH3 ChIP-seq data to the CS-CAU genome into five regions (Fig. 3A). Regions 1, 3, and 5 represent significant CENH3 enriched regions in one, two, or all *T. urartu* and *T. monococcum* accessions. Region 2 represents functional centromeres formed in hexaploid wheat. Region 4 represents areas that are neither ancestral centromere regions nor functional centromeres formed in the hexaploid wheat (Fig. 3A). We calculated the insertion times of CRW in these five regions and found that Region 2 contains the youngest CRW insertions. In contrast, Regions 1, 3, and 5, which represent ancestral centromeric regions, contain relatively older CRW insertions (Fig. 3C). Based on these observations, we propose an evolutionary scenario for the AA subgenome centromeres (Fig. 3D). During the evolutionary process following polyploidization, a series of CRW invasion events occurred. These invasions May induce centromere structural variation and move some ancient centromeric repeats to adjacent pericentromeric regions, as seen with the division of ancient centromeric regions into regions 1, 3, and 5 by CRW invasions. This process caused CENH3 nucleosomes to oscillate between older and newer CRWs. This oscillation continues until a recently active CRW area exhibits a stronger ability to bind CENH3 nucleosomes (Fig. 3D; Additional file 1: Fig. S6). Taken together, this variation is likely resulted from the invasion of transposable elements into the AA subgenomic centromeres during and after the polyploidization process, disrupting the originally continuous CENH3 peaks into scattered regions.

### The evolutionary history of DD subgenomic centromeres

The DD Subgenome of hexaploid wheat originates from at least two independent hybridization events, with Lineage 2 (L2) being the primary donor and Lineage 1 (L1) showing the greatest genetic distance from the current DD subgenome of common wheat [62, 63]. Notably, leveraging the *Ae. tauschii* pangenome resources [64], we performed dot plot alignment analysis among the CS-DD subcentromeres and representative accessions of three evolutionary lineages (L1-L3). While centromeres on chromosome 4D stand out,

all other centromeres in both diploid *Ae. tauschii* and the hexaploid DD subgenome displayed considerable sequence similarity albeit with subtle structural variations, especially those from the L2 lineage (Additional file 1: Fig. S10). The 4D centromere's unique behavior might be due to a repositioning event to a neighboring location in common wheat [65]. Since the DD Subgenome of hexaploid wheat was the last one to be merged and the donor relationship was clear, it was possible to trace the evolution trajectory of DD Subgenome centromeres in detail. By categorizing and calculating the insertion time of 965 flLTR-RTs in CS-DD, we identified five clades of CRWs in the DD centromeres (Additional file 1: Fig. S11A). Our analysis showed that the invasion events of CRWs did not occur uniformly in the functional DD centromeric region during the transition from diploid to hexaploid and subsequent evolution (Additional file 1: Fig. S11B and S11C). In contrast, we found that the intense episodes of RTs burst events were mainly driven by the more active CRW1&2 and CRW3, occurring between 0 and 0.5 Ma and reaching their peaks at 0.1 Ma (Additional file 1: Fig. S11C). We summarized the distribution of CRWs in the centromeric region from Cen1D to Cen7D and found that Cen1D and Cen4D were significantly different from other centromeres, with the copy number of CRW5 being much higher than that of the other CRWs, indicating that these two centromeres are paleo-centromeres (Table S1). The detailed analysis of the centromere evolution of DD subgenome is graphically summarized as followed.

### Cen1D and Cen4D are paleo-centromeres

The functional centromere skeletons of Cen1D and Cen4D are mainly composed of the older CRW5 (Additional file 1: Fig. S11B and Additional file 2: Table S1). The extended CENH3 binding region (Region 2) of centromere 1D in CS (CS-Cen1D) relative to that of *Ae. tauschii* is primarily composed of CRW1&2, indicating a relatively late insertion (Fig. 4A and 4C). This indicates that the expanded functional centromeric regions were newly formed due to CRW invasion, rather than utilizing the pre-existing pericentromeric region, which, as mentioned later, is older than the centromere (Additional file 1: Fig. S12). Additionally, the pericentromeric region has a low content of flLTR-RTs, primarily consisting of elements also found in chromosomal arms, such as RLG_famc1 (*Fatima*), RLG_famc2 (*Sabrina Egug*), RLC_famc1 (*Angela WIS*), and RLG_famc10 (*Carmilla Ifis*) (Fig. 4A) [47]. To compare the evolution trajectory of Cen1D in diploid and hexaploid levels, we calculated the insertion time of CRWs in Cen1D of *Ae. tauschii* (DD-Cen1D) and CS-Cen1D. Notably, DD-Cen1D is younger than CS-Cen1D (Fig. 4D). On the basis of the original skeleton, a significant number of CRWs in DD-Cen1D inserted around 0.1 Ma have been preserved. The small peak of about 1 Ma is the burst time of the original centromere skeleton, and the Main peak is at 0.1 Ma (Fig. 4E). In other words, a large amount of CRWs inserted into Cen1D was preserved 0.1 Ma, which is the reason for the expansion of *Ae. tauschii* Cen1D (Fig. 4B and 4E). The younger DD-Cen1D maintains centromeric function on the basis of ancient CRW5 by retaining continuous invasion of CRW1&2 at random (Fig. 4B and 4E). Additionally, there was a difference in the nucleotide level between the CRWs continuously introduced by DD-Cen1D and the expanded CRWs in CS-Cen1D. This led to some corresponding obvious continuous peaks not being detected in *Ae. tauschii* subcentromeric domains when the

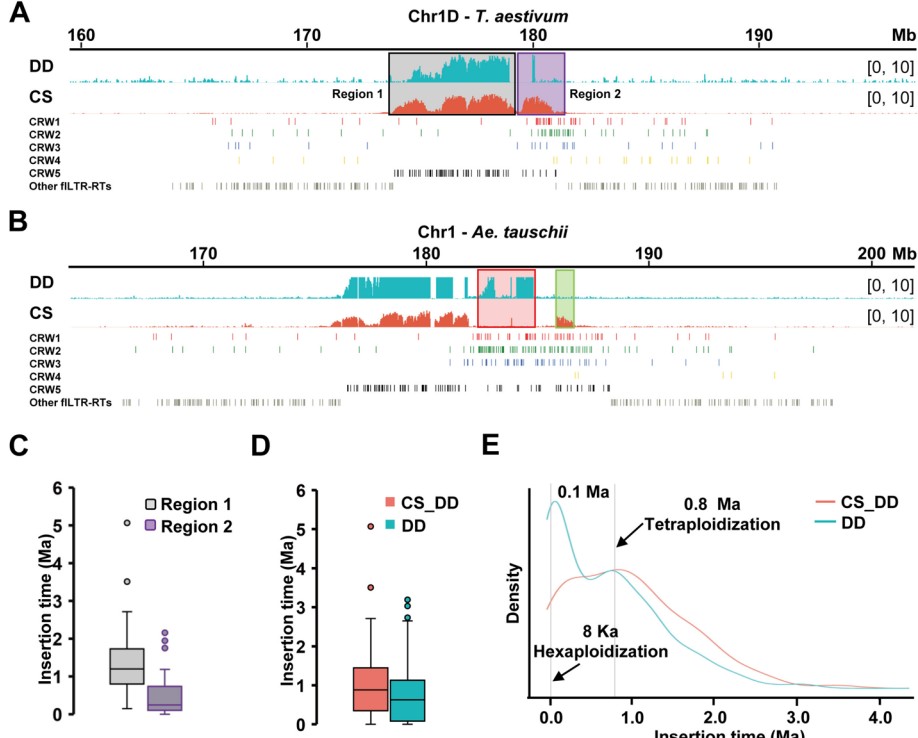

**Fig. 4** The evolutionary trajectory of Cen1D. **A-B** The detailed layout of the CENH3 enrichment [log₂(ChIP/Input)] ratio in centromeric regions of *T. aestivum* CS (**A**) and *Ae. tauschii* DD (**B**). *Ae. tauschii* (accession Y2282) is shown in cyan; CS in orange. The distribution of CRW1 (red), CRW2 (green), CRW3 (blue), CRW4 (yellow), CRW5 (black), and other flLTR-RTs (gray) in pericentromeres is shown in the respective tracks. **C** The insertion time of CRWs within chromosome 1D Subregions 1 and 2 in CS. The median insertion time in Region 1 is 1.20 Ma, and 0.23 Ma in Region 2. **D,E** Box plot (**D**) and kernel density estimate plot (**E**) of CRW insertion time in *Ae. tauschii* cen1D (cyan) and *T. aestivum* cen1D (orange). Two polyploid events are marked by dashed lines

CS CENH3-ChIP data were mapped to the *Ae. tauschii* reference genome, while some obscure peaks were detected in the adjacent area (Fig. 4B, red and green boxes).

CS-Cen4D, another paleo-centromere [34], is almost exclusively composed of CRW5 (Fig. 5A). The insertion times of these CRWs are much older than centromeric domains when the *Ae. tauschii* CENH3-ChIP data is Mapped to CS chromosome 4 (DD-CS-Cen4D) and DD-Cen4D (Fig. 5A and 5B), indicating the presence of epialleles-like centromeres in CS-Cen4D that differ in strength. Phylogenetic analysis shows a clear separation between CRWs in DD-CS-Cen4D and CS-Cen4D (Fig. 5C). The 2.24-Mb region of DD-CS-Cen4D was formed by intermittent insertion within a mix of non-centromere-specific TE relicts and other repetitive sequences during 0.1–0.2 Ma (Fig. 5A and 5D). However, DD-Cen4D evolved by the insertion of a large number of CRWs into DD-CS-Cen4D on the basis of retaining the original CRW5 skeleton to form a functional centromere of 4 Mb (Fig. 5A). Moreover, the CRWs in DD-Cen4D and DD-CS-Cen4D exhibited a highly similar region with a unique base-pair level ratio of over 2 Mb that could not be distinguished.

Additionally, several waves of transpositional bursts were observed in the pericentromere of chromosome 4D among 11 different types of LTR-RTs (Fig. 5D). This indicates that LTR-RTs in (peri)centromeres have undergone rapid independent evolution

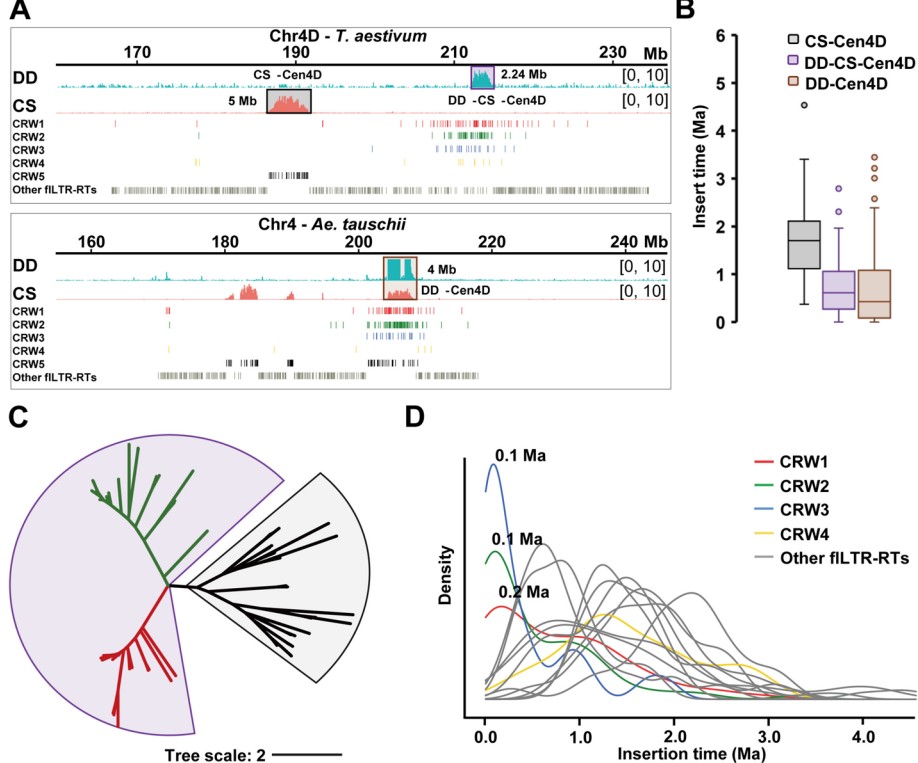

**Fig. 5** The evolutionary trajectory of Cen4D. **A** The detailed layout of the CENH3 enrichment [log$_2$(ChIP/Input)] ratio in the centromeric regions of CS (above) and *Ae. tauschii* (below). *Ae. tauschii* (accession Y2282) is shown in cyan; CS in orange. The distribution of CRW1 (red), CRW2 (green), CRW3 (blue), CRW4 (yellow), CRW5 (black), and other flLTR-RTs (gray) in pericentromeres is shown at their respective tracks. **B** The insertion time of CRWs associated with *Ae. tauschii* CENH3-ChIP samples and CS CENH3-ChIP samples in corresponding chromosome 4D subregions, analyzed by boxplot. The enriched region observed from mapping CENH3-ChIP samples of *Ae. tauschii* to the CS reference genome (DD-CS-Cen4D) is highlighted in violet, while the centromere 4D in *Ae. tauschii* (DD-Cen4D) is marked in brown. **C** RT phylogenetic relationship of the CRWs in DD-CS-Cen4D and CS-Cen4D. CRWs from DD-CS-Cen4D are shown in red and green, and CRWs from CS-Cen4D are shown in black. **D** Kernel density estimate plot of the time when flLTR-RTs inserted in the 4D pericentromere of *T. aestivum*. CRW1 is shown in red, CRW2 in green, CRW3 in blue, CRW4 in yellow, and all other types of flLTR-RTs in gray

by increasing their copy numbers after speciation. Moreover, from the aspect of insertion time, it is found that the activity degree of the pericentromere is much lower than that of the centromeric region (Fig. 5D).

### Genetic trajectories of Cen2D, Cen3D, Cen5D, Cen6D, and Cen7D

The functional centromeres of Chr2D, Chr3D, Chr5D, Chr6D, and Chr7D are primarily constructed by younger CRW1&2. The observed expansion of CenD to one or both sides, particularly the significant expansion seen in Cen6D, suggests a pattern that may be consistent with adaptation to a larger genome size (Fig. 6A and 6C). The older expanded region existed before polyploid formation, indicating they are not a newly inserted sequence (Fig. 6A and 6B). Additionally, certain pericentromeres flanking the centromere function as functional centromeres, enabling CENH3 loading post-polyploid formation, which accounts for the bimodal pattern seen in Fig. 2E (non-major peaks marked by blue arrows). Therefore, we propose an evolutionary model for DD

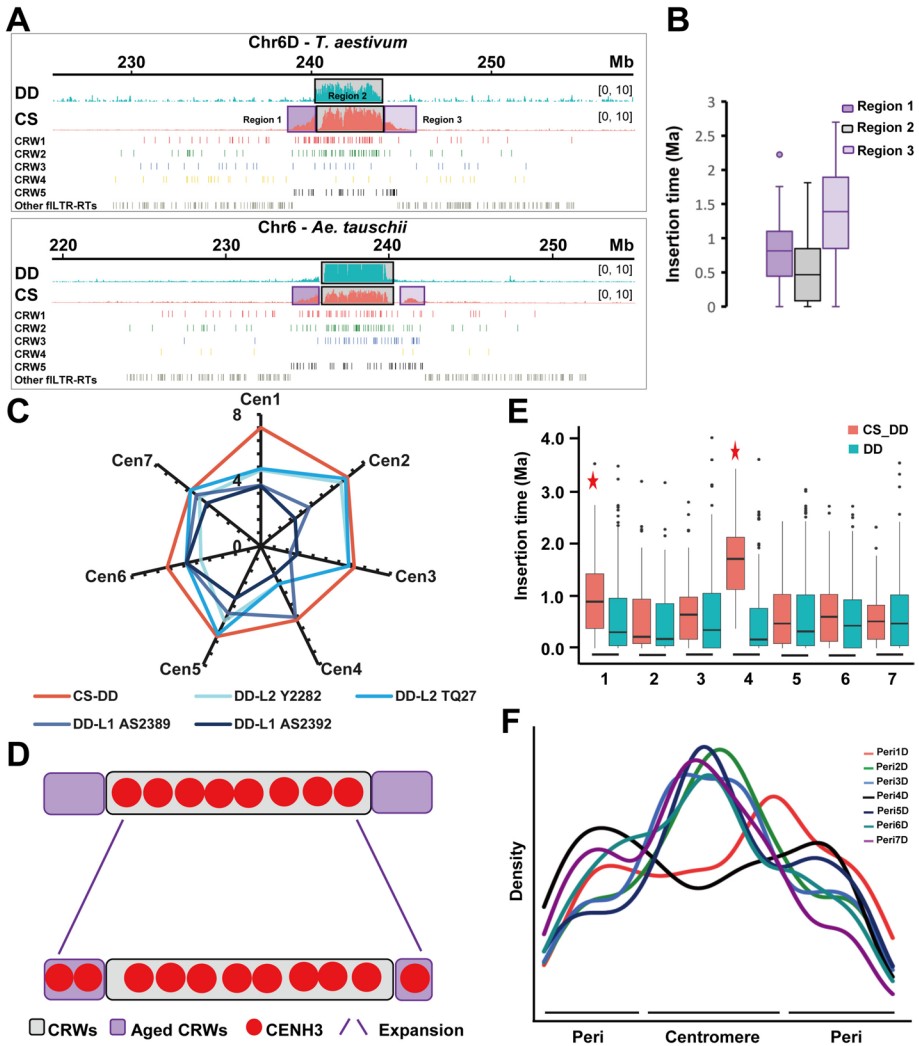

**Fig. 6** Evolutionary model of DD sub-centromeres in common wheat. **A** The detailed layout of the CENH3 enrichment [log$_2$(ChIP/Input)] ratio in the centromeric regions of CS (above) and *Ae. tauschii* (below). *Ae. tauschii* (accession Y2282) is shown in cyan; CS in orange. The distribution of CRW1 (red), CRW2 (green), CRW3 (blue), CRW4 (yellow), CRW5 (black), and other flLTR-RTs (gray) in pericentromeres is shown at the following tracks, respectively. **B** The insertion time of CRWs associated with *Ae. tauschii* CENH3-ChIP samples and CS CENH3-ChIP samples in corresponding chromosome 6D subregions, analyzed by boxplot. **C** Relative plot showing the difference in size between CENH3-binding region of four *Ae. tauschii* (cyan) and *T. aestivum* (orange) among Chr1-7D. The axis represents centromere size (megabases). The light color represents *Ae. tauschii* from the L2 lineage, the dark color represents the L1 lineage, and different accessions are marked after the lineage name. **D** Dynamics of DD subgenome centromeres in allohexaploid wheat. Expansion arrows (purple) indicate epigenetic expansion of CENH3 domains into pericentromeres (purple hatched areas: ancestral CRW regions). Partial CENH3 occupancy (red circles) in ancestral pericentromeric regions reflects functional plasticity, facilitating adaptation to genomic complexity (see bimodal CENH3 distribution in Fig. 2E). **E** The box plot of CRW insertion time in *Ae. tauschii* (cyan) and *T. aestivum* (orange) among homoeologous groups Chr1 to Chr7. **F** Kernel density estimate plot of flLTR-RTs along DD sub-pericentromeres of *T. aestivum* in homoeologous groups Chr1 to Chr7. The pericentromere 1D (Peri1D) is red, Peri2D is green, Peri3D is blue, Peri4D is black, Peri5D is navy blue, Peri6D is cyan, and Peri7D is purple

subgenomic functional centromeres in young allohexaploid common wheat, where CENH3 exhibiting a tendency to extend towards the pericentromere relative to diploid *Ae. tauschii* (Fig. 6D).

Furthermore, aside from the ancient centromeres 1D and 4D, the density of flLTR-RTs in the other five functional centromere regions was unequivocally higher than in the pericentromeric regions (Fig. 6E–6F). This may serve as an important genetic marker of functional centromeres at the DNA level, alongside the CENH3 epigenetic marker, which is crucial for centromere function establishment and maintenance. Based on these findings, we suggest that CENH3-binding domains may be demarcated by changes in the overall distribution density of flLTR-RTs. The dynamics of CRW loci and their distribution density variations in maintaining centromere function in the preliminary stage of polyploid wheat evolution warrant further investigation.

### The DD subcentromeres did not expand in synthetic hexaploid wheat

To determine whether the DD subcentromeric CENH3 expansion is a revolutionary change at the onset of polyploid speciation (explosive expansion) or an evolutionary process throughout the entire evolutionary cycle (slow accumulation expansion), we synthesized hexaploid wheat Za-Y2 (*T. turgidum* ssp. *dicoccoides* × *Ae. tauschii*, BBAADD). This designed synthetic wheat system combines wild progenitor genomes with reference sequences, providing unique access to the initial centromere states following polyploidization that are normally inaccessible in natural populations. Technical replicates were performed, demonstrating good reproducibility of the ChIP-seq data (Additional file 1: Fig. S4C-S4D). The ChIP-seq reads of *Ae. tauschii* (Y2282), the second generation of synthesized allopolyploid ($S_2$), and the third generation of synthesized allopolyploid ($S_3$) were mapped back to the merged reference genome of the corresponding parents for analysis. Results showed that the $S_2$ and $S_3$ generations of synthetic hexaploid Za-Y2 could not reproduce the obvious CENH3 depositing trend toward the DD subgenomic pericentromere (Fig. 7A). In addition, the localization of DD subgenomic CENH3 nucleosomes during continuous self-crossing in $S_2$ and $S_3$ showed a strong positive correlation with *Ae. tauschii*, with $R^2$ reaching 0.91 (Fig. 7B and 7C). Therefore, it is speculated that the expansion of DD subcentromeric CENH3 in common wheat is not induced by hybridization and chromosome doubling events, but rather a relatively slow process, which is a gradual response.

### Discussion

#### Hypothesis of centromere evolution in AA subgenome

The discontinuous centromeric regions of *T. urartu* exhibited in the common wheat AA subgenome recorded unique evolutionary features resulting from two successive polyploidization events followed by long-term evolution (Fig. 3A; Additional file 1: Fig. S6). Three hypotheses attempt to explain these features, each with inherent challenges. The first hypothesis, which suggests *T. urartu* is not the direct donor of *T. aestivum*, contradicts substantial genomic evidence on wheat evolution [12, 31–33]. The second hypothesis posits that massive nested rearrangements akin to chromothripsis, might have occurred in the centromeric region [66]. However, this hypothesis is contradicted by both the absence of multiple translocations and inversions between chromosomes in successful long-established allopolyploids wheat, along with evidence that polyploid centromeres resist structural homogenization (e.g., suppressed intersubgenomic exchange in *Brachypodium* [15], and the striking collinear conservation

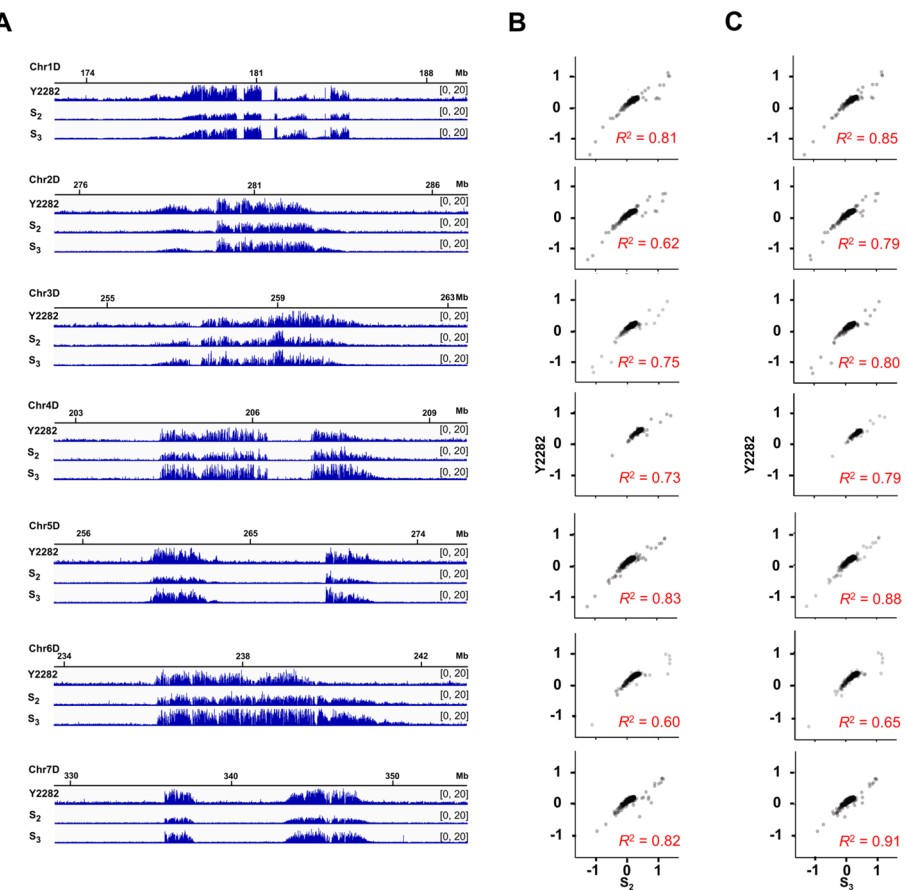

**Fig. 7** CENH3 nucleosomes showed limited deposition in pericentromeres of resynthesized hexaploid wheat. **A** The detailed layout of the CENH3 profile in *Ae. tauschii* (Chr1-7D). From top to bottom are shown as [log$_2$(ChIP/Input)] ratio data of paternal line Y2282, S$_2$, and S$_3$. **B, C** Correlation analysis. The centromeric localizations of CENH3 nucleosomes on the paternal *Ae. tauschii* Chr1-7 chromosomes showed a significant positive correlation between the paternal line Y2282 and S$_2$ (**B**), and the paternal line Y2282 and S$_3$ (**C**), respectively

of pericentromeres between *T. urartu* and the corresponding AA subgenome of *T. aestivum* (Fig. 3B; Additional file 1: Fig. S7). Furthermore, centromere fragmentation is generally considered an evolutionarily dead end [67, 68]. The third hypothesis suggests that other sequences may have invaded the core centromeric region, eventually replacing or partially replacing the original centromere sequence, leading to the formation of a new genetically functional centromeric region by successfully forming CENH3 nucleosomes (Fig. 3D). Evidence from Cen3B supports this hypothesis, showing that young LTR-RTs in the functional centromeric region have a higher affinity for CENH3 binding [29, 69]. This evolutionary pattern finds striking parallels in cotton polyploidization, where centromeres undergo extensive reorganization including CRC (centromere-associated repetitive sequences) homolog transfer between subgenomes [70, 71], and in *Arabidopsis* with *ATHILA* LTR-RTs [20, 21]. These findings suggest that retrotransposons are highly dynamic within centromeres during evolution, with the centromeres of AA and DD subgenomes in a continuous state of evolution (Fig. 6E; Additional file 1: Fig. S8D).

While centromeric DNA composition changes represent a key driver of centromere evolution during allopolyploidization, it is equally critical to consider the co-evolution of CENH3 variants. The expansion from two CENH3 variants in diploids to six in hexaploid wheat provides additional regulatory complexity that may facilitate functional specialization. Notably, diploid barley studies revealed that while two CENH3 variants (α and β) coexist, only αCENH3 maintains cross-species compatibility in wheat-barley hybrids, suggesting variant-specific loading constraints may influence centromere stability during genome merger [71]. In polyploid wheat, this complexity escalates—tetraploids show adaptive evolution of βCENH3 through positive selection [39]. These observations suggest that centromere evolution in allopolyploids operates through two interconnected mechanisms—dynamic DNA sequence turnover through retrotransposon activity and functional diversification of CENH3 variants. Future studies should explore how variant-specific CENH3 deposition patterns (particularly βCENH3's functional specialization) interact with chromatin architecture and spatial organization to maintain centromere stability during and after wheat genome mergers.

### Different fates of CRWs

The classification of centromeric LTR-RTs in common wheat varies across published literature, leading to inconsistencies. To address this, we analyzed sequences from multiple sources to align our findings with existing data [29, 47]. Our analyses suggest that within the *Gypsy* superfamily, the CRW1&2 correspond to two distinct but related RLG_famc8.1 and RLG_famc8.3, respectively, while CRW3 aligns with RLG_famc8.2, and CRW4, also known as *Weg*, corresponds to RLG_famc39 [29, 47]. Distinct evolutionary patterns among CRW subfamilies (Additional file 1: Fig. S12) suggests CRW3 may represent a relatively inactive lineage that has remained stable throughout polyploidization events, potentially due to its critical structural or functional role in centromere organization. In contrast, CRW1&2 appear to have undergone more active species-specific evolution, possibly through adaptive diversification or transpositional bursts following genome merger or duplication.

The evolutionary trajectories of these CRW lineages differ markedly. CRW4 and CRW5 experienced bursts of amplification followed by decline and near-extinction (Additional file 1: Fig. S12A, S12C, and S12E). CRW1&2 may follow a similar path trajectory over time. The fate of CRW3 remains uncertain, as it is the sole non-autonomous CRW within the centromere. Its strong sequence conservation between diploid and hexaploid wheat (Additional file 1: Fig. S9) suggests either an essential structural role (analogous to the noaCRR1 subfamily in rice [49]) or stringent functional constraints limiting its evolutionary plasticity. The evolutionary events of CRWs, involving their invasion, degeneration, and survival within functional centromeres, gain significance in an evolutionary context particularly when they take place in the germline, ensuring transmission to future generations [72, 73]. Retrotransposons, prone to deletions and mutations, rarely remain intact over time [74]. Full-length elements indicate recent insertions, while aged elements exhibit truncations, nested insertions, and mutations leading to degeneration. Both intact and truncated CRWs accumulate in pericentromeres, with CRW1&2 being predominant, though traces of CRW3 and CRW4 remained (Additional file 1: Figs. S3 and S12).

In Gramineae, it is common for multiple CR subfamilies to share functional centromeres [28, 29, 36, 48, 49], leading to competition and inconsistent content among subfamilies. Different subfamilies take on central roles at various evolutionary stages, ensuring that centromeres remain functional and avoid quiescence by involving multiple CR families [28, 29, 36, 48, 49]. For instance, CRW4 may have originally served as the skeleton of functional centromeres in wheat but was eventually replaced by CRW1&2 over evolutionary time (Additional file 1: Fig. S12). The subfamily-specific conservation of CRW1-3 (Additional file 1: Fig. S9) underscores their collective yet temporally staggered contributions to centromere maintenance, with each lineage potentially compensating for the decline of others. As ancient CRW copies are progressively purged (Additional file 1: Fig. S3), CRW4 appears to be nearing extinction or assimilation into CRW5.

### Peri-centromere is a candidate for functional centromere

This study offers new insights into the establishment and maintenance of functional centromeres, revealing that both the pericentromeric and centromeric regions have the potential to establish functional centromeres (Fig. 6D). However, subtle differences in their capacity to maintain these structures are evident. The pericentromere, which may be a vestige of a former centromere, ultimately lost the tug-of-war with the centromere, rending it less competitive. In hexaploid wheat, for example, the expanded centromeric region of CenD traces back to the pericentromeric region of its diploid progenitor, *Ae. tauschii*, rather than to recently invading CRWs (Fig. 6A and 6D). This suggests that the pericentromere is a viable candidate for functional centromere formation. Furthermore, the presence of satellite sequences and CRWs outside the CENH3-associated region implies that while these elements may contribute to the nucleation of the CENH3 binding domain, they do not strictly define centromeric boundaries [75]. An extreme example is the maize B chromosome, which harbors large clusters of centromere DNA elements that are distant from the functional centromere [76].

These findings support the idea that Ty3/*gypsy*-like retrotransposons are integral to grass centromeres and that the CENH3 loading onto CR elements is likely region-specific rather than sequence-specific (Fig. 6F; Additional file 1: Fig. S8A-S8C). This highlights the importance of considering not only DNA sequences but also two- and three-dimensional epigenetic information, such as non-B-form DNA structures [77–81], in maintaining centromere functions, including the definition of boundaries. Additionally, the blurring of pericentromere and centromere boundaries over evolutionary time aligns with the extension of evolutionary timelines observed for CenA and CenD (Fig. 3A; Additional file 1: Fig. S6 and S12).

### Newly visited functional centromeres are highly malleable in allopolyploids

The dynamic depositing of CENH3 nucleosomes to flanking regions of CRW clusters appears crucial for appropriately increasing centromere size in relatively small, newly introduced chromosomes, such as those from *Ae. tauschii* into the nucleus of *T. turgidum*'s. This CENH3 deposition in pericentromeric regions likely helps balance the increased spindle traction during anaphase, thereby preventing lagging chromosomes in a larger nucleus [82, 83]. Notably, our study involved synthesizing hexaploid wheat

mimic and self-crossing them to the third generation to track the evolution of the DD subgenomic centromere. We observed the CENH3 loading dynamics remained stable from speciation to the third generation in these resynthesized polyploidies, suggesting a gradual centromeric response to allopolyploid speciation. This gradual evolution of centromere CENH3 in synthetic hexaploid wheat may underlie the mechanism ensuring genome stability in allopolyploid, laying the foundation for the subsequent centromere evolution [84].

Interestingly, *T. aestivum* and *Ae. tauschii*, chromosome 4D remains dicentric, with difference in the inactivation process: the native centromere is inactivated in *Ae. tauschii*, while the de novo centromere is inactivated in *T. aestivum*. This suggests that the de novo centromere in *Ae. tauschii* is either as strong as or stronger than the native one, or the de novo one in *T. aestivum* is degraded (Fig. 5). Hybridizing these two special chromosomes into a single nucleus allows for the observation of offspring transmissibility. This system presents a unique opportunity to investigate the molecular composition and stoichiometry of both stronger and weaker centromeres, as well as to test specific models of centromere strength. The mechanism behind the rapid positional switch of centromere activity, which is necessary to avoid chromosome instability due to transient acentric or dicentric states, remains unclear. The loss of CENH3 from the original position in the native centromere of *Ae. tauschii* and in the de novo centromere of *T. aestivum*, respectively, is particularly puzzling and warrants further investigation.

### Young CRWs maintain optimal transcription levels or transcripts for function centromeres

Maintaining an optimal level of transcription or transcript levels within the chromatin environment is essential for centromere function, a balance achieved by precise loading of CENH3 proteins [85–90]. Over evolutionary time, young CRWs invade the core centromeric region (Fig. 3; Additional file 1: Fig. S12), accumulating truncations, nested insertions, and mutations that degrade the original flLTR-RTs. This degradation leads to the loss of promoter-like sequences in LTRs, rendering them transcriptionally inactive and compromising their ability to maintain functional centromeres [91–94]. However, during the reverse transcription cycle, LTR sequence are regenerated, resulting in intact and complete sequences that can initiate transcription [95]. The competition between young and aged CRWs in maintaining centromere function leads to the gradual replacement of degraded centromere sequences, forming a new centromere island (Fig. 3). Our findings suggest that young flLTR-RTs compensate for the reduced transcriptional activity of aged CRWs, preserving the specific chromatin environment essential for centromere function. The evolution of wheat centromeres is driven by the opposing forces of incipient and senescent CRWs, with deletion and mutation acting as continuous forces eroding the centromeric chromatin environment. When degradation reaches a certain threshold, especially under harsh environmental conditions, bursts of LTR-RT activity, such as CRW1&2 in *Triticum*, pAWRC1 in *Secale* [50], may occur. In these instances, young flLTR-RTs replace or partially replace the functional centromere sequences, compensating for the degraded centromeric region pushing the degraded LTR-RTs to one or both sides and preserving the specific centromere chromatin environment essential for centromere function. Notably, even truncated CRWs, despite their

degradation, continue to act as backbone of the centromere during evolution (Additional file 1: Fig. S3 and S12).

The relationship between maintaining functional centromeres and the continuous invasion of CRWs remains a subject of debate. While our hypothesis suggests a correlation between these factor, it does not imply causality, nor does it exclude other factors, such as CENH3 protein, in driving the centromere evolution [19]. A counterargument is that we assumed a constant rate of functional centromere degeneration (flLTR-RTs mutation rate), while retrotransposon bursts are ubiquitous across flowering plants [26]. These findings enhance our understanding of the multiple patterns and stages of functional centromere evolution in allopolyploid subgenomes and suggest that differences in centromere characteristic, or the timing and sequence of their integration into the same nucleus, may result in distinct evolutionary trajectories. Further research is needed to explore the potential involvement of unknown factors, such as unique chromatin structures and spatial interaction, in the maintenance of functional centromeres.

In conclusion, this study reconstructs a dynamic evolutionary atlas of centromere plasticity in polyploid wheat, revealing distinct evolutionary trajectories among subgenomic centromeres mediated by CRW activity. Our integrated analysis demonstrates that wheat centromeres function as genomic archives recording the dynamic interplay between CRWs, where the recent expansion of CRW1&2 subfamilies lineages (Fig. 3; Additional file 1: Figs. S3 and S12) coexists with the conserved architecture of CRW3 (Additional file 1: Fig. S9). Comparative analysis of synthetic and natural wheat lineages reveals remarkable subgenome divergence: AA centromeres experienced recurrent CRW invasions while DD centromeres exhibited gradual epigenetic expansion (Figs. 3, 6, and 7; Additional file 1: Figs. S7 and S10), highlighting the complex evolutionary dynamics of centromere restructuring during polyploidization. By bridging pangenome diversity with comparative genomics, these findings provide fundamental insights into the mechanistic basis of subgenomic centromere evolution and establish a valuable framework for potential crop improvement strategies.

## Methods

### Plant materials

The hybrids of Zavitan × Y2282 (*T. turgidum* ssp. *dicoccoides* × *Ae. tauschii*, BBAADD) were generated in our laboratory following a previously described embryo rescue technique [84]. All other materials used in this study, including diploid, tetraploid, and hexaploid species from the *Triticum* and *Aegilops* genera, are listed in Table 2. We chose representative lines covering diploid, tetraploid, and hexaploid wheat to reflect the polyploidization process, prioritizing accessions with published reference genomes to ensure accurate centromere comparisons [3, 33, 35, 37, 38, 58, 59, 96]. For allohexaploid, we selected CS, AK58 (with high-quality genome and ChIP data [59, 60]), and TAA10 (an allohexaploid bread wheat line used to produce extracted tetraploid genomes for polyploidization studies [61, 96]); for tetraploids, we included both wild (Zavitan) and cultivated (Svevo) types with high-quality reference genome assemblies [3, 58]; and for diploids, we chose *T. urartu* G1812 and *Ae. tauschii* Y2282 [33, 36]. Additional accessions like *T. monococcum* (accessions KU104, KT003-002, TA299, and TA10622) [35, 51, 52], *T. urartu* (accession TMU38 and TMU06) [53, 54], and *Ae. tauschii* (accessions

TA10171, TA1675, TA2576, and TQ27) [38, 55–57] were included to represent geographical diversity and cross-compatibility. The synthetic hexaploid *T. turgidum* × *Ae. tauschii* (Za-Y2) was specifically selected as a wild-derived synthetic wheat with unique genetic background that provides novel insights into polyploidization mechanisms. Seeds were germinated at room temperature (25 °C) until the buds and root tips were 2 to 3 cm long, after which the plants were transplanted into soil and grown under long-day conditions (16 °C) in a growth chamber for 20 days.

### CENH3 chromatin immunoprecipitation sequencing and ChIP-seq analysis

The ChIP-seq experiments were performed using wheat-specific anti-αCENH3 antibodies targeting the conserved peptide ARTKHPAVRKTK-C, which recognizes all αCENH3 variants across A/B/D subgenomes [39]. This antibody selection was based on its consistent centromere labeling throughout mitotic phases and established use in wheat centromere studies [12, 35]. The experiments followed a previously described method [97], with ChIP sequencing performed on the NovaSeq system (Illumina NovaSeq 6000), generating paired-end reads with at least $1 \times$ genome coverage. Mapping of ChIP-seq reads to the wheat reference genome was carried out as described in Su et al. [11].

### Identification and characterization of putative flLTR-RTs within centromere and pericentromere

The quality of anti-CENH3 ChIP-seq and Input-seq reads were assessed, filtered, and analyzed using latest high-quality assembled genomes as reference genomes, following the method described in Su et al. [11]. The positions of functional CENH3-binding regions are strictly defined by ChIP data and input data, with the pericentromeric regions being 15 Mb on each side of AA and BB Subgenome functional centromeres, and 10 Mb on each side of DD subgenome functional centromeres. The sequence of the centromere and pericentromere was extracted using BEDTools [98]. The flLTR-RTs were identified using LTR_retriever [44], where LTR_finder [43] and LTRharvest [45] were employed to identify flLTR-RTs based on the principle of structure ab initio. The following parameter settings were used: "ltr_finder -D 25000 -d 3000 -L 2000 -l 100 -p 20 -C -M 0.85" and "ltrharvest -overlaps best -seed 30 -minlenltr 100 -maxlenltr 2000 -mindistltr 3000 -maxdistltr 25,000 -similar 85 -mintsd 4 -maxtsd 20 -motif tgca -motifmis 1 -vic 60 -xdrop 5 -mat 2 -mis −2 -ins −3 -del −3," respectively. LTR_retriever was then used to integrate both candidates.

### LTR retrotransposon insertion time

The date of flLTR-RT insertions was based on the divergence between the 5b and 3a LTR sequences using the emboss distmat, applying the Kimura two-parameter correction [99]. Molecular dating of flLTR-RT insertion ages was estimated using the formula: age = distance/2r, where r is the average Substitution rate of $1.3 \times 10^{-8}$ per site per year [24, 47, 100].

### *Phylogenetic analysis*

To account for the presence of frame-shift mutations and stop codons in ancient retrotransposon elements, multiple alignments of all non-overlapping flLTR-RTs candidates

were construct using Clustal W (implemented in MEGA 7.0), and a Maximum Likelihood tree was generated from the alignments using IQtree Maximum Likelihood method with bootstrap values selected for 1000 replications (IQ-TREE multicore, version 2.0.3) [101, 102]. To further customize the phylogenetic tree, Interactive Tree Of Life (iTOL, version 6.8) was used [103]. The LTR-RTs were then classified into different subfamilies based on classification criteria defined by Wicker et al. [46]. It is important to note that partially ambiguous CRWs, particularly those located at boundaries, were manually calibrated using the Basic Local Alignment Search Tool (BLAST) online (https://blast.ncbi.nlm.nih.gov/).

### Characterization of diversity within centromeric repeats in T. aestivum genomes

The distribution of sequence identity of all satellite monomers to the CentT552 or CentT566 consensus was generated from the LASTZ results. The comparisons of repeat sequence similarity within each sub/genome were performed as previously reported [15]. The dot plot of the centromere comparison between different sub/genomes or self-comparison was performed using the Redotable (https://www.bioinformatics.babraham.ac.uk/projects/redotable/).

### *Fluorescence in situ hybridization*

Young root tip cells and meiosis cells from various plants were collected for FISH as described previously [11]. CRWs were labeled with Alexa Fluor-594-5-dUTP (red) or Alexa Fluor-488-dUTP (green), using probes prepared by PCR amplification with primer pairs Listed in Additional file 2: Table S2. Chromosome samples from different wheat lines were subjected to the same experimental conditions, including equal amounts of probes and the same hybridization time and temperature. FISH images were obtained by confocal microscopy (Cell Observer spinning disk confocal microscope, Zeiss) using identical exposure time and processed with Photoshop CS 6.0 (Adobe).

## Supplementary Information

Additional file 1. This file contains Figures S1-S12.

Additional file 2. This file contains Tables S1-S2.

Additional file 3. Review history.

### Acknowledgements

We thank Nathan D. Han from The Edison Family Center and James A. Birchler from University of Missouri for critical reading of the article and helpful comments. We thank the high-performance computing platform at National Key Laboratory of Crop Genetic Improvement in Huazhong Agricultural University.

### Peer review information

### Authors' contributions

Y.H.H., H.D.S. and F.P.H. designed the experiments and the initial version of the paper. Y.H.H. synthesized hexaploid wheat, performed the anti-CENH3 ChIP-seq experiment, and wrote the manuscript. Y.H.H., Y.L., and C.L. performed data analysis. H.D.S., Y.H.H., Y.L., C.L., F.P.H., C.Y.Y., J.S.L., H.Q.L. and F.P.H. contributed to the revisions. All authors contributed to subsequent versions and have read and approved the manuscript. Y.H.H., Y.L., and C.L. contributed equally.

## Funding

This project was financially supported by the National Natural Science Foundation of China (31991212 and 32400451) and the National Key Research and Development Program of China (2022YFF1003303).

## Data availability

ChIP-seq for all Triticeae tribe species used in this study have been deposited in the NGDC Genome Sequence Archive (GSA; https://bigd.big.ac.cn/gsa/) under accession number CRA011849 and project number PRJCA018324 [104]. The representative consensus sequences of CRWs can be found in the GenBank/EMBL database under the following accession numbers: CRW1&2, OR344338; CRW3, OR344339. The FISH data have been deposited in the Figshare [105]. The Chinese Spring assembly is available at NGDC under project number PRJCA018511 [37].

## Declarations

### Ethics approval and consent to participate
Not applicable.

### Competing interests
The authors declare no competing interests. Fangpu Han is a guest editor of the Centromere Structure and Evolution article collection, but was not involved in the editorial decision making or peer review of this manuscript.

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

## 