## [Additional file 3. Review history. · Genome Biology]

Dear editor:

On behalf of my co-authors, we thank you very much for giving us an opportunity to revise our manuscript. Also, we appreciate the editor and reviewers very much for their positive and constructive comments and suggestions on our manuscript entitled “Distinct evolutionary trajectories of subgenomic centromeres in polyploid wheat”.

We have studied the reviewers’ comments carefully and tried our best to made revisions thoroughly, which are clearly marked in the paper. We have highlighted the novelty in the results and discussion sections. In addition, we have expanded our analyses by integrating existing data from domesticated einkorn and wild einkorn genomes and also the pan-genome of *Ae. tauschii*. Moreover, we have provided a dedicated explanation to justify the observed discrepancies with previous studies in the results and discussion sections. We have considered each comment and provided appropriate explanations or modifications accordingly. Please find our point-by-point responses below.

We would like to express our sincere appreciation to you and reviewers for the insightful comments on our paper. Looking forward to hearing from you.

Reviewer Comments:

Reviewer reports:

Reviewer #1:

The authors presented a well-designed and well-written study on the likely evolution of the wheat centromeres. The data are new and interesting.

Reply: We thank the reviewer for the valuable and constructive comments and suggestions.

Line 62

Could you please explain better what you mean by “a whole set of smaller centromeres with smaller chromosomes”?

Reply: Thank you for your suggestion. We have revised this sentence to make the intended meaning more clearly. “*T. turgidum* then merged with *Ae. tauschii*, whose chromosomes are notably smaller than those of the A/B genomes, to form the modern common wheat fewer than 10,000 years ago.”

Line 104

Please provide the reference for the wheat T2T reference genome.

Reply: Thank you. We have added the citation of reference in the manuscript.

Reference:

1. Wang Z, Miao L, Tan K, Guo W, Xin B, Appels R, Jia J, Lai J, Lu F, Ni Z, et al: Near-complete assembly and comprehensive annotation of the wheat Chinese Spring genome. *Mol Plant* 2025, 18:892-907.

Line 116 Considering that hexaploidy wheat encodes at least six variants of CENH3, specify whether the applied CENH3 antibody is recognizing all CENH3 variants of wheat.

Reply: Thank you for your suggestion. Hexaploid wheat encodes six CENH3 variants (all located on homoeologous group 1 chromosomes 1A/1B/1D). Multiple sequence alignment confirmed that the divergence of amino acids in the N-terminal of α CENH3 and β CENH3 protein (Fig. 1). The peptides used to generate specific α CENH3 and β CENH3 antibodies (highlighted in red and green boxes, respectively) are different, which were confirmed in the previous report [1]. The immunofluorescence signals demonstrated that the different incorporation of α CENH3 and β CENH3 within centromeres throughout mitotic cell cycles, and α CENH3 displays stably labels all the cell cycles in wheat (Fig. 2). As anti- β CENH3 antibody localized centromeres by immunofluorescence, we also performed CENH3-ChIP-seq experiment several times but no enrichment of CENH3, which suggests that this antibody may not be suitable for ChIP-seq applications. Given α CENH3 antibody have been using in several different wheat centromere studies (e.g., [2, 3]), we employed specific CENH3-ChIP-seq with anti- α CENH3 antibody in our manuscript. We added this explanation in the method sections.

Fig. 1 Multiple sequence alignment results of α CENH3 and β CENH3.

Fig. 2 Distribution of α CENH3 and β CENH3 in CS root tip cells. α CENH3 signals are red, β CENH3 signals are green, and DAPI-stained chromosomes are blue. Bars = 10 μ m.

Reference:

1. Yuan J, Guo X, Hu J, Lv Z, Han F: Characterization of two CENH3 genes and their roles in wheat evolution. *New Phytol* 2015, 206:839-851.
2. (IWGSC) IWGSC: Shifting the limits in wheat research and breeding using a fully annotated reference genome. *Science* 2018, 361:eaar7191.
3. Ahmed HI, Heuberger M, Schoen A, Koo DH, Quiroz-Chavez J, Adhikari L, Raupp J, Cauet S, Rodde N, Cravero C, et al: Einkorn genomics sheds light on history of the oldest domesticated wheat. *Nature* 2023, 620:830-838.

Line 123

The statement that centromere size is an independent feature that does not correlate with the size of the genome is too strong; because it is only based on three subgenomes, and the opposite conclusion has been reported multiple times before. See for example: <https://www.nature.com/articles/s41598-021-99386-7> Thus, reconsider your conclusion.

Reply: Thank you for your suggestion. We have revised the statement to better reflect the relationship between centromere size and genome size: “Centromere sizes in this system do not follow the subgenome size hierarchy (BB > AA > DD). This might be due to our limited samples

of only three subgenomes, which does not invalidate the broader pattern across eukaryotes [1].”
We have revised it in the manuscript.

Reference:

1. Plackova K, Bures P, Zedek F: Centromere size scales with genome size across eukaryotes. *Sci Rep* 2021, 11:19811.

Line 196

Please provide the reference for the *T. urartu* reference genome.

Reply: Thank you. We have added the citation of the *T. urartu* reference genome in the manuscript.

Fig. 3c. Please extend the legend describing the evolutionary scenario for the AA subgenome centromere. Explain the symbol of expansion and CENH3, as shown in Figure 6D.

Reply: Thank you for your suggestion. We have revised the figure legends for these figures to clarify the evolutionary dynamics of centromeres, respectively.

Fig. 3D (AA subgenome): Episodic invasions by CRWs (black arrows) displace ancestral centromeric repeats to pericentromeric domains, driving dynamic redistribution of CENH3 nucleosomes between ancestral CRW clusters (blue shading) and emerging CRW arrays (green shading). Epigenetic competition culminates in dominant CENH3 occupancy at nascent retrotransposon arrays, while relocating ancient CRWs to unilateral or bilateral pericentromeric regions. Through continuous integration of novel CRW insertions and reorganization of ancestral repeats, this process establishes an evolutionarily dynamic CRW cluster architecture that ultimately shapes the functional centromeres of the AA subgenome.

Fig. 6D (DD subgenome): Dynamics of DD subgenome centromeres in allohexaploid wheat. Expansion arrows (purple) indicate epigenetic expansion of CENH3 domains into pericentromeres (purple hatched areas: ancestral CRW regions). Partial CENH3 occupancy (red circles) in ancestral pericentromeric regions reflects functional plasticity, facilitating adaptation to genomic complexity.

Do you propose that during and after allopolyploidization, the organization of the centromere differs?

Reply: Thank you for your suggestion. We actually propose that the organization of the centromere differs during and after allopolyploidization. It is well-established that allopolyploidization triggers localized genomic reorganization including transposable element (TE) activation and gene loss during initial stages of wheat [1, 2]. Recently studies in centromere evolution of cotton polyploidization, they observed centromeres undergo extensive reorganization including size expansion, repositioning and structure variations occurred post-polyploidization. CRC (Centromere-associated repetitive sequences) homologs were transferred from the diploid DD subgenome progenitor to the AA subgenome, invaded the AA subgenome and then underwent post-tetraploidization proliferation [3, 4]. Similarly, substantial centromeric sequence and structural variation without inter-subgenomic exchange were observed from diploid to tetraploid in *Brachypodium* genus [5]. While our study did not detect pronounced centromeric structural changes in early polyploid generations, we observed that the potential subtle differences, such as TE mobilization events and significant variation in centromeric sequence architecture size in established allohexaploid wheat. These discoveries provide novel perspectives on chromosomal evolution in polyploid systems. We have incorporated this discussion into the discussion section of the manuscript.

Reference:

- 1 (IWGSC) IWGSC: Shifting the limits in wheat research and breeding using a fully annotated reference genome. *Science* 2018, 361:eaar7191.
2. Session AM, Rokhsar DS: Transposon signatures of allopolyploid genome evolution. *Nat Commun* 2023, 14:3180.
3. Hu G, Wang Z, Tian Z, Wang K, Ji G, Wang X, Zhang X, Yang Z, Liu X, Niu R, et al: A telomere-to-telomere genome assembly of cotton provides insights into centromere evolution and short-season adaptation. *Nat Genet* 2025, 57:1031-1043.
4. Yan H, Han J, Jin S, Han Z, Si Z, Yan S, Xuan L, Yu G, Guan X, Fang L, et al: Post-polyploidization centromere evolution in cotton. *Nat Genet* 2025, 57:1021-1030.
5. Chen C, Wu S, Sun Y, Zhou J, Chen Y, Zhang J, Birchler JA, Han F, Yang N, Su H: Three

near-complete genome assemblies reveal substantial centromere dynamics from diploid to tetraploid in *Brachypodium* genus. *Genome Biol* 2024, 25:63.

Line 320

I like the described experiment.

Reply: Thank you for your supportive comment.

Line 341 and others

Discussion, hypothesis of centromere evolution

I would like to suggest an additional hypothesis. So far, the authors only considered the centromeric DNA composition changes in the allopolyploidization process. But besides this process, also, the complexity of available CENH3 variants increased. Diploids only encode two CENH3 variants, and hexaploid express up to six CENH3s. Could this new level of CENH3 complexity also influence the centromere organization?

Reply: We appreciate for your constructive comments. We strongly agree that the expansion of CENH3 variants constitutes a fundamental aspect of centromere evolution in allopolyploidization process. Previous studies in diploid barley (*Hordeum vulgare*), while two CENH3 variants (α and β) are encoded, only α CENH3 could be incorporated into wheat centromeres in wheat-barley addition lines [1]. Our studies in polyploid wheat have demonstrated that β CENH3 has undergone adaptive evolution through positive selection and elevated expression [2], and revealed cell cycle-dependent localization patterns across six CENH3 variants in common wheat, particularly β CENH3 interphase-specific enrichment, indicative of functional specialization during polyploid evolution [2]. These findings collectively support the hypothesis that CENH3 variant diversification enhances regulatory plasticity, working synergistically with centromeric DNA changes to fine-tune centromere architecture and ensure chromosomal stability in polyploids. For future investigations, generating β CENH3 antibodies suitable for CHIP experiments would be valuable to further elucidate its molecular functions. We have incorporated this perspective into the Discussion section.

Reference:

1. Sanei M, Pickering R, Kumke K, Nasuda S, Houben A: Loss of centromeric histone H3 (CENH3) from centromeres precedes uniparental chromosome elimination in interspecific barley hybrids. *Proc Natl Acad Sci U S A* 2011, 108.
2. Yuan J, Guo X, Hu J, Lv Z, Han F: Characterization of two CENH3 genes and their roles in wheat evolution. *New Phytol* 2015, 206:839-851.

Figure 8 shows the same as Figures 3c and 6d. Either remove Figure 8 or prepare a new figure.

Reply: Thank you for your suggestion. We have removed Figure 8.

#Reviewer 2

The manuscript by Huang et al. presents a robust analysis of centromeres in bread wheat. It produces lots of interesting datasets, but it lacks any novel findings. This manuscript presents some unique understanding about centromeric landscapes to some extent, but it does not provide any solid evidence in support of these indicative findings. The present form does not present any unique and sound scientific findings that can warrant its publication in *Genome Biology*.

Reply: Thank you for your suggestion. We have thoroughly revised the manuscript to highlight the key findings that distinguish our study from previous work on centromere evolution during polyploidization.

Our study integrates a comprehensive ploidy series of *Triticum* and *Aegilops* species, including wild and cultivated diploids, tetraploids, and hexaploid wheat, along with a near-complete assembly of common wheat (*T. aestivum*). This system provides an unprecedented window into centromere dynamics during polyploidization. Previous centromere studies in polyploids (e.g., cotton, oat, *Nicotiana benthamiana*, and *Brachypodium*) primarily focused on allotetraploids and lacked comparisons with progenitor species [1-6]. For example, recent work in cotton (*Gossypium hirsutum* cv. “Zhongmian 113”) revealed post-polyploidization centromere reorganization, including size expansion, repositioning, and structural variations, as well as the invasion and proliferation of DD-subgenome-derived centromeric repeats into the AA-subgenome [1-2]. However, such studies typically analyzed only the polyploid itself without ancestral comparisons. In contrast, our study employs multiple accessions per ploidy level (diploid, tetraploid, and hexaploid) alongside high-quality reference genomes, enabling systematic dissection of centromere evolution across the entire wheat polyploidization trajectory.

Furthermore, we incorporated additional high-quality reference genomic resources, including complete assembly of centromere in *T. monococcum* (A^mA^m) [7] and a comprehensive pan-

genome of *Ae. tauschii* (DD) [8]. Our analysis further validated the high assembly quality of centromeres in these reference genomes. These resources facilitate cross-ancestral comparisons and provide further evidence in support of these unique understanding about centromeric landscapes during polyploidization.

We also included the newly synthesized hexaploid wheat (S_2 – S_3 generations) derived from wild *T. dicoccoides* × *Ae. tauschii* crosses (both parents with reference genomes). By analyzing centromere dynamics in early-generation synthetics alongside long-term stabilized polyploids (e.g., CS, TAA10, and AK58), we reveal initial centromere reorganization post-polyploidization and long-term evolutionary stabilization mechanisms. The integration of these diverse datasets collectively reflects the differential responses of centromeres in subgenomes during both the early and long-term evolutionary processes of wheat polyploidization. We have added these points into the manuscript to better emphasize the novelty and mechanistic contributions of our work.

Reference:

1. Hu G, Wang Z, Tian Z, Wang K, Ji G, Wang X, Zhang X, Yang Z, Liu X, Niu R, et al: A telomere-to-telomere genome assembly of cotton provides insights into centromere evolution and short-season adaptation. *Nat Genet* 2025, 57:1031-1043.
2. Yan H, Han J, Jin S, Han Z, Si Z, Yan S, Xuan L, Yu G, Guan X, Fang L, et al: Post-polyploidization centromere evolution in cotton. *Nat Genet* 2025, 57:1021-1030.
3. Li W, Wang Y, Liu J, He Q, Zhou Y, Li M, Liu N, Liang H, Yun Y, Gong Z, Du H: A gap-free complete genome assembly of oat and OatOmics, a multi-omics database. *Mol Plant* 2025, 18:179-182.
4. He Q, Li W, Miao Y, Wang Y, Liu N, Liu J, Li T, Xiao Y, Zhang H, Wang Y, et al: The near-complete genome assembly of hexaploid wild oat reveals its genome evolution and divergence with cultivated oats. *Nat Plants* 2024, 10:2062-2078.
5. Chen C, Wu S, Sun Y, Zhou J, Chen Y, Zhang J, Birchler JA, Han F, Yang N, Su H: Three near-complete genome assemblies reveal substantial centromere dynamics from diploid to tetraploid in *Brachypodium* genus. *Genome Biol* 2024, 25:63.
6. Chen W, Yan M, Chen S, Sun J, Wang J, Meng D, Li J, Zhang L, Guo L: The complete genome assembly of *Nicotiana benthamiana* reveals the genetic and epigenetic landscape of centromeres. *Nat Plants* 2024, 10:1928-1943.

7. Ahmed HI, Heuberger M, Schoen A, Koo DH, Quiroz-Chavez J, Adhikari L, Raupp J, Cauet S, Rodde N, Cravero C, et al: Einkorn genomics sheds light on history of the oldest domesticated wheat. *Nature* 2023, 620:830-838.
8. Cavalet-Giorsa E, González-Muñoz A, Athiyannan N, Holden S, Salhi A, Gardener C, Quiroz-Chávez J, Rustamova SM, Elkot AF, Patpour M, et al: Origin and evolution of the bread wheat D genome. *Nature* 2024, 633:848-855.

Some specific points:

1. In the abstract, they claim to reveal that the wheat centromere contains 5 CRWs, which is already established by several independent studies. I agree with their presentation that they have looked at the landscape of centromere adaptation and evolution, but this analysis is very superficial.

Reply: Thank you for your suggestion. Based on previous studies of five CRW types classification [1], we further systematically analyzed the landscape of centromere adaptation and evolution based on wheat near-complete assembly reference genome data. We believe it is essential to first establish these fundamental data, which serve as the basis for our entire research. Based on this foundation, we further integrated genetic materials covering multiple ploidy levels (both wild and cultivated) and cross-species pan-genome datasets. These analyses revealed novel findings, including distinct evolutionary patterns among CRW subfamilies (Additional file 1: Fig. S8). Specifically, CRW3 elements formed tightly conserved clusters between wild einkorn (TA299) and hexaploid wheat (CS), while CRW1&2 showed clear divergence between these species. This differential pattern suggests CRW3 may represent a relatively stable lineage maintained throughout polyploidization events, potentially due to its critical structural or functional role in centromere organization. In contrast, CRW1&2 appear to have undergone more active species-specific evolution, likely through adaptive diversification or transpositional bursts following genome merger or duplication. We have added these results in the manuscript.

Reference:

1. Wicker T, Gundlach H, Spannagl M, Uauy C, Borrill P, Ramirez-Gonzalez RH, De Oliveira R, International Wheat Genome Sequencing C, Mayer KFX, Paux E, Choulet F: Impact of transposable elements on genome structure and evolution in bread wheat. *Genome Biol* 2018, 19:103.

2. Interestingly, they claim that they have uncovered a series of CRW invasion events that have shaped the evolution of AA subgenome centromeres. I agree that it may lead to it, but no direct data presented led them to such a firm conclusion. I would love to see a thorough, standalone comparison of the recently published einkorn genomes and A-genomes included in this study. Ahmed et al have reported highly dynamic centromeres in A-genome and provided a nice study where whole centromeres can be assembled. I am unsure why this data was not compared, as everything is available in the public domain. Another study from Heuberger et al. (2024) demonstrates the role of two transposable elements and suggests that the LTR elements recruit and/or phase new CENH3 deposition. They argue that the interplay between the TE families and the plant host dynamically maintains wheat centromeres. I do not see any reference to these two important studies in the manuscript, and these studies indicate that A-genome centromeres are dynamic and may not be in the same line as supported by this study.

Reply: Thank you for your suggestion. We have added these two studies as citations in our manuscript. First, we conducted comprehensive comparisons between *T. monococcum* and CS genomes with near-complete assembly of centromeres. We integrated high-quality genomes of domesticated einkorn (*T. monococcum* TA10622) and wild einkorn (*T. monococcum* TA299) [1], and systematically compared the centromeric sequences of *T. monococcum*, *T. urartu*, and AA subgenome of CS wheat using dot plots (Figs. 3B, S7). The dot plot analysis revealed that the CENH3-binding core regions showed high sequence similarity between the two *T. monococcum* accessions and *T. urartu*. Significantly higher conservation was observed within *T. monococcum* accessions than between *T. monococcum* and *T. urartu* accessions. After allopolyploidization, substantial sequence changes were detected between the AA-subgenome centromeres of common wheat and those of diploid *T. monococcum* and *T. urartu*, while *T. urartu* showed relatively smaller divergence from CS AA-subgenome centromeres. These results demonstrate significant changes in AA-subgenome centromeric sequences during polyploidization

Then, we further performed comparative analysis of CRW insertion timing using these reference genomes (Fig. S8D). The CRW insertion times in wild einkorn (*T. monococcum* TA299) centromeres were significantly younger than those in *T. urartu* and CS-AA. For chromosomes 1-5, CS-AA showed younger CRW insertion times than *T. urartu*, while the opposite trend was observed for chromosomes 6-7. These results suggest that *T. monococcum* centromeres may have undergone more recent TE-mediated remodeling, and that after

allopolyploidization, CRW retrotransposons in most AA-subgenome chromosomes experienced recent bursts of activity, with potential chromosome-specific differences.

Finally, following Heuberger et al. [2], we systematically classified CRWs in wild einkorn and CS (Fig. S9). CRW1&2 (*Cereba* subfamily) formed distinct evolutionary clusters between diploid and hexaploid centromeres, while CRW3 (*Quinta* subfamily) showed tight clustering across ploidy levels (CRW4 was excluded due to low abundance). These differential patterns support complex dynamic evolutionary mechanisms in AA-subgenome centromeres during allopolyploidization. These results have been incorporated into the main text.

Fig. 3 (B) Dot plot alignments (300-bp window) of pericentromeric and centromeric regions for chromosomes 4A, comparing *T. monococcum* (domesticated einkorn TA10622; wild einkorn TA299), *T. urartu* (G1812), and *T. aestivum* (CS).

Fig. S8 (D) Comparison of CRWs insertion time in *T. urartu* (orange), *T. monococcum* (blue), and *T. aestivum* (cyan) among homoeologous groups Chr1 to Chr7.

Fig. S9 Phylogenetic analysis of filTR-RTs from centromeric regions of *T. monococcum* (TA299) and *T. aestivum* (CS-AA).

Reference:

1. Ahmed HI, Heuberger M, Schoen A, Koo DH, Quiroz-Chavez J, Adhikari L, Raupp J, Cauet S, Rodde N, Cravero C, et al: Einkorn genomics sheds light on history of the oldest domesticated wheat. *Nature* 2023, 620:830-838.
2. Heuberger M, Koo D-H, Ahmed HI, Tiwari VK, Abrouk M, Poland J, Krattinger SG, Wicker T: Evolution of Einkorn wheat centromeres is driven by the mutualistic interplay of two LTR retrotransposons. *Mobile DNA* 2024, 15.
3. The sample sizes selected to perform such an analysis are small, and some variations observed in the centromeric regions may be due to assembly-related problems. Sup fig 5 kind of explains it. As the authors explained in the method section, ChIP-seq reads were mapped at very low depth and did not rule out presence-absence variations due to lack of coverage. It needs to be clarified. Since Einkorn centromeres presented by Ahmed et al and Heuberger et al, 2024 show a complete structural analysis based on deep sequencing data, authors must include those accessions for this comparative analysis, and I am sure it will clarify some of the questions that may come out due to the lack of coverage of assembly errors.

Reply: Thank you for your suggestion. We have reassessed the quality of our ChIP-seq datasets, confirming stable 10-15 fold enrichment in centromeric regions across all samples. These sequencing depths are sufficient to exclude coverage-related artifacts in observed presence-absence variations (PAVs). Importantly, we have incorporated the Einkorn centromeric CENH3-ChIP data from Ahmed et al. [1] into our analysis (Figs. 3A, S5, S6), which demonstrates consistent PAV detection patterns when aligned to both *T. urartu* (Fig. S5) and *T. aestivum* genomes (Figs. 3A, S6). Our cross-genome alignment analyses reveal that reciprocal mappings of CENH3-ChIP data to *T. monococcum*, *T. urartu* and *T. aestivum* genomes (Fig. S6) systematically detect more PAVs in cross-species comparisons. These results consistent with our dot plot results and supporting the biological basis of these variations. These analyses were included in the revised manuscript, and demonstrate that the observed centromeric variations reflect biological differences rather than technical artifacts related to assembly or coverage limitations.

Fig. 3 (A) The detailed layout of the CENH3 enrichment [$\log_2(\text{ChIP}/\text{Input})$] ratio along the pericentromeric region of bread wheat chromosome 4A.

Fig. S5 Genome-wide mapping of CENH3 ChIP-seq reads from four *T. monococcum* accessions (KT003-002, KU104, TA299, and TA10622) and three different *T. urartu* accessions (G1812, TMU06, and TMU38) to the *T. urartu* reference genome.

Fig. S6 Diploid wheat exhibits a sporadic distribution pattern along the *T. aestivum* AA pericentromeres. The density distribution of CENH3 [$\log_2(\text{ChIP}/\text{Input})$] along the chromosomes of the AA subgenome of hexaploid wheat *T. aestivum*.

Reference:

1. Ahmed HI, Heuberger M, Schoen A, Koo DH, Quiroz-Chavez J, Adhikari L, Raupp J, Cauet S, Rodde N, Cravero C, et al: Einkorn genomics sheds light on history of the oldest domesticated wheat. *Nature* 2023, 620:830-838.

4. Similar comparisons should also be made through the D-genome using *Ae. tauschii* by utilizing the pangenome data that was recently made available through Emile Cavalet-Giorsa et al. 2024, which was published in Nature. I am not sure if visible significant variations in *T. urartu* are an artifact, as it is probably based on the draft genome. It also needs to be clarified in the revised version.

Reply: Thank you for your suggestion. We have now incorporated the *Ae. tauschii* pan-genome data [1] into our study. Although CENH3-ChIP data are not publicly available for the TA10171 (L1), TA1675 (L2), and TA2576 (L3) accessions, we mapped the existing ChIP-seq data from Y2282 line (L2 lineage) to the AY61 reference genome (another L2 lineage accession) [2], which revealed well-defined CENH3 enriched domain. These results suggested that the high quality of centromere assembly of these *Ae. tauschii* genomes. To further investigate

centromere dynamics, we performed comparative dot-plot analyses between the DD-subgenome centromeres of *T. aestivum* CS and the *Ae. tauschii* pangenomes (new Fig. S10). The results showed that substantial sequence conservation among *Ae. tauschii* centromeres, albeit with subtle structural variations. We also detected minor structural differences between *Ae. tauschii* and the CS DD-subgenome centromeres, while observing relatively strong conservation between the L2 lineage (TA1675) and CS, with minimal structural divergence. These findings highlight the dynamic nature of centromeres at both pangenome and polyploidization levels. These results have been integrated into the manuscript (Fig. S10).

Fig. S10 Comparative analysis of D-genome centromere architecture between *Ae. tauschii* and *T. aestivum*. (Chromosome 1 shown; see supplementary figures for other chromosomes).

Reference:

1. Cavalet-Giorsa E, González-Muñoz A, Athiyannan N, Holden S, Salhi A, Gardener C, Quiroz-Chávez J, Rustamova SM, Elkot AF, Patpour M, et al: Origin and evolution of the bread wheat D genome. *Nature* 2024, 633:848-855.
2. Zhou Y, Bai S, Li H, Sun G, Zhang D, Ma F, Zhao X, Nie F, Li J, Chen L, et al: Introgressing the *Aegilops tauschii* genome into wheat as a basis for cereal improvement. *Nat Plants* 2021, 7:774-786

5. I am unsure of the importance of the information provided in supplementary figures 3 and 8. I can understand the point for the sup fig 3, but failed to understand the rationale behind the sup fig8.

Reply: Thank you for your suggestion. Fig. S3 was retained because it provides experimental validation of CRW localization patterns across wheat ploidy levels, which is essential for supporting our conclusions on centromere evolution. We have removed the Fig. S8 from the final manuscript.

6. Authors should clearly explain their generated data and the datasets from publicly available resources. In the method section, they refer to Table 1. However, it does not explain why these lines were selected, whether they were chosen as a unique representative of genetic diversity or for other specific reasons. CS, Svevo, and Zavitan make sense, but what about other accessions? How can a person from a non-wheat background depict it?

Reply: Thank you for your suggestion. We have chosen representative lines covering diploid, tetraploid, and hexaploid wheat to reflect the polyploidization process, utilizing accessions with published reference genomes to ensure accurate centromere comparisons [1-8]. For allohexaploid, we selected CS, AK58 (with high-quality genome and ChIP data [8]), and TAA10 (an allohexaploid bread wheat line used to produce extracted tetraploid genomes for polyploidization studies [3, 9]); for tetraploids, we included both wild (Zavitan) and cultivated (Svevo) types with high-quality reference genome assemblies [5-6]; and for diploids, we chose *T. urartu* G1812 and *Ae. tauschii* Y2282 [4, 10]. Additional accessions like *T. monococcum* (accessions KU104, KT003-002, TA299, and TA10622) [1, 11, 12], *T. urartu* (accession

TMU38 and TMU06) [13-14], and *Ae. tauschii* (accessions TA10171, TA1675, TA2576, TQ27, As2392, and As2389) [2, 15-17] were included to represent geographical diversity and cross-compatibility. The synthetic hexaploid *T. turgidum* × *Ae. tauschii* (Za-Y2) was specifically selected as a wild-derived synthetic wheat with unique genetic background that provides novel insights into polyploidization mechanisms. We have clarified these criteria in the Methods section and revised Table 2 by adding "ChIP-data source".

Table 2. Species and datasets used in this study (public data and current study).

Species and cultivars	Accession	Genome formula	ChIP-data source
Diploid ($2n = 2x = 14$)			
T. monococcum	KU104 [11]	A ^m A ^m	This study
T. monococcum	KT003-002 [12]	A ^m A ^m	This study
T. monococcum	TA299	A ^m A ^m	Ahmed et al., 2024 [1]
T. monococcum	TA10622	A ^m A ^m	Ahmed et al., 2024 [1]
T. urartu	G1812 [4]	A ^u A ^u	This study
T. urartu	TMU06 [13-14]	A ^u A ^u	This study
T. urartu	TMU38 [13-14]	A ^u A ^u	This study
Ae. tauschii	Y2282 [10]	DD	This study
Ae. tauschii	TA10171	DD	Cavalet-Giorsa et al., 2024 [2]
Ae. tauschii	TA1675	DD	Cavalet-Giorsa et al., 2024 [2]
Ae. tauschii	TA2576	DD	Cavalet-Giorsa et al., 2024 [2]
Ae. tauschii	TQ27 [15]	DD	This study
Ae. tauschii	As2392 [16]	DD	This study
Ae. tauschii	As2389 [17]	DD	This study
Tetraploid ($2n = 4x = 28$)			
T. turgidum ssp. dicoccoides	Zavitan [5]	BBA ^u A ^u	This study
T. turgidum ssp. durum	Svevo [6]	BBA ^u A ^u	This study
Hexaploid ($2n = 6x = 42$)			
T. aestivum	CS [7]	BBA ^u A ^u DD	This study
T. aestivum	AK58 [8]	BBA ^u A ^u DD	Liu et al., 2024 [18]
T. aestivum	TAA10 [9]	BBA ^u A ^u DD	This study
T. turgidum (Zavitan) × Ae. tauschii (Y2282)	Za-Y2	BBA ^u A ^u DD	This study

Reference:

1. Ahmed HI, Heuberger M, Schoen A, Koo DH, Quiroz-Chavez J, Adhikari L, Raupp J, Cauet S, Rodde N, Cravero C, et al: Einkorn genomics sheds light on history of the oldest domesticated wheat. *Nature* 2023, 620:830-838.
2. Cavalet-Giorsa E, González-Muñoz A, Athiyannan N, Holden S, Salhi A, Gardener C, Quiroz-Chávez J, Rustamova SM, Elkot AF, Patpour M, et al: Origin and evolution of the bread wheat D genome. *Nature* 2024, 633:848-855.
3. Kerber ER: Wheat: Reconstitution of the tetraploid component (AABB) of hexaploids. *Science* 1964, 143:253-255.
4. Ling HQ, Ma B, Shi X, Liu H, Dong L, Sun H, Cao Y, Gao Q, Zheng S, Li Y, et al: Genome sequence of the progenitor of wheat A subgenome *Triticum urartu*. *Nature* 2018, 557:424-428.

5. Avni R, Nave M, Barad O, Baruch K, Twardziok SO, Gundlach H, Hale I, Mascher M, Spannagl M, Wiebe K, et al: Wild emmer genome architecture and diversity elucidate wheat evolution and domestication. *Science* 2017, 357:93-97.
6. Maccaferri M, Harris NS, Twardziok SO, Pasam RK, Gundlach H, Spannagl M, Ormanbekova D, Lux T, Prade VM, Milner SG, et al: Durum wheat genome highlights past domestication signatures and future improvement targets. *Nat Genet* 2019, 51:885-895.
7. Wang Z, Miao L, Tan K, Guo W, Xin B, Appels R, Jia J, Lai J, Lu F, Ni Z, et al: Near-complete assembly and comprehensive annotation of the wheat Chinese Spring genome. *Mol Plant* 2025, 18:892-907.
8. Jia J, Zhao G, Li D, Wang K, Kong C, Deng P, Yan X, Zhang X, Lu Z, Xu S, et al: Genome resources for the elite bread wheat cultivar Aikang 58 and mining of elite homeologous haplotypes for accelerating wheat improvement. *Mol Plant* 2023, 16:1893-1910.
9. Zhang H, Zhu B, Qi B, Gou X, Dong Y, Xu C, Zhang B, Huang W, Liu C, Wang X, et al: Evolution of the BBAA component of bread wheat during its history at the allohexaploid level. *Plant Cell* 2014, 26:2761-2776.
10. Zhao J, Xie Y, Kong C, Lu Z, Jia H, Ma Z, Zhang Y, Cui D, Ru Z, Wang Y, et al: Centromere repositioning and shifts in wheat evolution. *Plant Commun* 2023, 4:100556.
11. Murai K, Nishiura A, Kazama Y, Abe T: A large-scale mutant panel in wheat developed using heavy-ion beam mutagenesis and its application to genetic research. *Nucl Instrum. Methods Phys Res B* 2013, 314:59-62.
12. Huang Y, Liu Y, Guo X, Fan C, Yi C, Shi Q, Su H, Liu C, Yuan J, Liu D, et al: New insights on the evolution of nucleolar dominance in newly resynthesized hexaploid wheat *Triticum zhukovskiyi*. *Plant J* 2023, 115:1298-1315.
13. Guo X, Han F: Asymmetric epigenetic modification and elimination of rDNA sequences by polyploidization in wheat. *Plant Cell* 2014, 26:4311-4327.
14. Shcherban AB, Khlestkina EK, Sergeeva EM, Salina EA: Genomic changes at early stages of formation of allopolyploid *Aegilops longissima* × *Triticum urartu*. *Genetika* 2007, 43:963-970.
15. Khasdan V, Yaakov B, Kraitshtein Z, Kashkush K: Developmental timing of DNA elimination following allopolyploidization in wheat. *Genetics* 2010, 185:387-390.
16. Liu F, Si H, Wang C, Sun G, Zhou E, Chen C, Ma C: Molecular evolution of *Wcor15* gene enhanced our understanding of the origin of A, B and D genomes in *Triticum aestivum*. *Sci Rep* 2016, 6:31706.
17. Lin H, Shi T, Zhang Y, He C, Zhang Q, Mo Z, Pan W, Nie X: Genome-wide identification, expression and evolution analysis of m6A writers, readers and erasers in *Aegilops tauschii*. *Plants (Basel)* 2023, 12.
18. Liu C, Huang Y, Guo X, Yi C, Liu Q, Zhang K, Zhu C, Liu Y, Han F: Young

retrotransposons and non-B DNA structures promote the establishment of dominant rye centromere in the 1RS.1BL fused centromere. *New Phytol* 2024, 241:607-622.

Dear editor:

On behalf of my co-authors, we sincerely appreciate the reviewers' time and constructive feedback, which has significantly improved our manuscript. Below we provide point-by-point responses to all comments.

Reviewer 1

I'm satisfied with the revised version of the manuscript.

Reply: We thank the reviewer for their positive assessment of our revisions.

Reviewer 2

Authors have done an excellent job in addressing the concerns we had in the manuscript and revising the manuscript. I am happy with the revisions and feel that the manuscript can be accepted.

Reply: We are grateful to the reviewer for acknowledging our efforts in addressing the previous concerns. We are pleased that the revised version now meets the journal's standards.

We believe the manuscript has been strengthened through this revision process and hope it is now suitable for publication in Genome Biology.